# Pioneer and repressive functions of p63 during zebrafish embryonic ectoderm specification

José M. Santos-Pereira [1], Lourdes Gallardo-Fuentes [1], Ana Neto[1], Rafael D. Acemel [1] & Juan J. Tena [1]

The transcription factor p63 is a master regulator of ectoderm development. Although previous studies show that p63 triggers epidermal differentiation in vitro, the roles of p63 in developing embryos remain poorly understood. Here, we use zebrafish embryos to analyze in vivo how p63 regulates gene expression during development. We generate *tp63*-knock-out mutants that recapitulate human phenotypes and show down-regulated epidermal gene expression. Following p63-binding dynamics, we find two distinct functions clearly separated in space and time. During early development, p63 binds enhancers associated to neural genes, limiting Sox3 binding and reducing neural gene expression. Indeed, we show that p63 and Sox3 are co-expressed in the neural plate border. On the other hand, p63 acts as a pioneer factor by binding non-accessible chromatin at epidermal enhancers, promoting their opening and epidermal gene expression in later developmental stages. Therefore, our results suggest that p63 regulates cell fate decisions during vertebrate ectoderm specification.

[1] Centro Andaluz de Biología del Desarrollo (CABD), Consejo Superior de Investigaciones Científicas/Universidad Pablo de Olavide, 41013 Seville, Spain. Correspondence and requests for materials should be addressed to J.J.T. (email: jjtenagu@upo.es)

The transcription factor (TF) p63 belongs to the p53 tumor-suppressor protein family and is a master regulator of ectoderm development that is known to play a key role during epidermal specification[1,2]. Its heterozygous mutations in humans have been associated with severe hereditary malformations affecting different ectoderm-derived structures, including skin, palate, digits, hair, and mammary glands[3–7]. These defects correspond to three main phenotypes: split-hand/foot malformation, ectodermal dysplasia, and orofacial cleft[8]. Interestingly, loss-of-function animal models, including mutant mice and knocked-down zebrafish, recapitulate the main human phenotypes, showing absence of skin, limbs and defects in palate development[9–12].

Recently, the role of p63 over enhancers and genes associated to epidermis differentiation has been highlighted[13–18], as well as its interaction with chromatin remodeling factors and epigenetic regulators[14,19–23]. The involvement of p63 in enhancing chromatin accessibility in epidermal cells has also been suggested[14,16], similarly to its related TF p53[24]. According to these observations, p63 is a likely candidate to act as a pioneer TF, which are TFs able to bind to non-accessible sites and open them by displacing nucleosomes, alone or assisted by other factors, in order to allow the binding of non-pioneer TFs and other proteins[25–28]. Pioneer activity is essential for establishing an active state of chromatin at regulatory regions characterized by nucleosome modifications as H3K4me1 and H3K27ac. However, a pioneer activity of p63 has not been demonstrated so far.

In addition to its role promoting epidermal development, p63 has been involved in the inhibition of other cell fates. In this regard, p63-null mice show an up-regulation of genes required for mesoderm development[29,30], and zebrafish gastrulating embryos knocked-down for p63 showed an expansion of the neural domains, while p63 over-expression leads to its reduction[11,31]. Interestingly, this work proposed p63 as the neural repressor postulated by the neural default model, according to which ectodermal cells become neural unless inhibited by BMP signaling[32], and p63 expression is induced by BMP signaling[11,18,31]. However, the mechanism by which p63 might inhibit neuroectoderm expansion is completely unknown.

In this work, we use zebrafish embryos as a model to gain insight into the molecular mechanisms by which p63 regulates the expression of its target genes. For this, we generate a p63-null zebrafish mutant using CRISPR/Cas9 technology and combine it with the integration of transcriptomic, genomic and epigenomic data. In particular, we use ChIPmentation to analyze the chromatin binding dynamics of p63 during zebrafish development and find two different functions of p63. During early development, p63 binds to enhancers associated to neural plate-expressing genes, where it limits Sox3 binding and neural gene expression. Indeed, p63 and Sox3 are co-expressed at the neural plate border. On the other hand, p63 binds enhancers associated to epidermis-expressing genes when they are in a non-accessible chromatin state, leading to its opening and epidermal gene expression. Therefore, our results suggest that p63 is an important regulator of ectoderm specification in vertebrates that plays a dual role as a repressor of the neural fate and as a pioneer TF that promotes the epidermal commitment.

## Results

**Deregulated epidermal expression in *tp63*<sup>−/−</sup> zebrafish mutant**. In order to study the effects of the lack of p63 in vertebrate ectoderm development, we generated a *tp63* knockout zebrafish model using the CRISPR/Cas9 genome editing system[33]. For this, we targeted the exon 3 of the *ΔNp63* isoforms (exons 5 or 6 of the *TAp63* isoforms), generating a 4-bp deletion that lead

to a premature stop codon within that exon that affects all *tp63* isoforms, leading to the absence of p63 protein in mutant animals (Fig. 1a). *tp63*<sup>−/−</sup> embryos died just after hatching, between 40 and 50 h post-fertilization (hpf), and from 36 hpf they showed defects in ectoderm-derived structures, including skin, pectoral fin buds and the fin fold, as reported previously in humans[4,6,7]. From this stage, *tp63*<sup>−/−</sup> mutant embryos could be unambiguously identified by observing the lack of pectoral fin buds and the reduced size of the fin fold, while heterozygous *tp63*<sup>+/−</sup> embryos showed wild-type phenotype (Fig. 1a, b). In *tp63*<sup>−/−</sup> embryos, in situ hybridization of *tbx5a* shows the formation of the mesodermal anlage that will generate the fin bud, but the apical ectodermal ridge (AER) is not formed and therefore the appendage does not grow (Fig. 1b). Although they were able to grow until more than 40 hpf and pigment cells were less organized but normally visible, the epidermis stopped developing and protecting the animals. Indeed, they died commonly because of a generalized attack by microorganisms or, in a few cases, by neural tissue extrusion. These phenotypes are similar to those described for *trp63* knockout mice, which die after birth due to dehydration and show craniofacial abnormalities, limb truncations as a result of failure of the AER to differentiate, and absence of epidermis and related appendages, including hair follicles, teeth and mammary glands[9,10]. Zebrafish embryos knocked-down for *tp63* also showed similar but milder phenotypes[11,12].

To study the impact of p63 loss at the transcriptomic level, we performed RNA-seq in wild-type and *tp63*<sup>−/−</sup> mutant embryos at 36 hpf. Differential gene expression analysis produced 973 up- and 1,371 down-regulated transcripts (Fig. 1c). The average fold-change of the down-regulated transcripts was significantly higher than that of the up-regulated ones (Fig. 1d), suggesting a higher global impact of p63 loss over activated genes than over repressed genes at this stage. Consistently with p63 functions, Gene Ontology term enrichment analyses showed biological processes related to epidermis and fin development for the down-regulated genes, and to mesoderm development for the up-regulated ones (Fig. 1e). Moreover, analysis of enrichment of WT expression patterns showed an over-representation of genes expressed in the epidermis for the down-regulated genes and in mesoderm tissues for the up-regulated ones (Fig. 1f). Altogether, these results agree with the positive role of p63 in epidermis and limb development[4–7,9,10] and with a proposed negative role over mesoderm development[29,30].

**Dynamic p63 binding during ectoderm specification**. In order to study in detail how p63 regulates its genetic network during development, we analyzed chromatin binding of p63 by ChIP-mentation (ChIP-seq coupled to Tn5-mediated TAGmentation of chromatin)[34] in different embryonic time-points: 80% of epiboly (8.3 hpf), 24 hpf and 36 hpf, corresponding to gastrulation, segmentation and pharyngula developmental stages, respectively (Fig. 2a, Supplementary Fig. 1). We found a total of 30,597 p63 binding sites (BSs) across the three stages. *K*-means clustering of these BSs produced four clusters (a, b, c and d), two of which showed a clear dynamic behavior: cluster b represented 3,956 early p63 BSs and cluster c was composed of 3,801 late p63 BSs (Fig. 2b). To elucidate whether dynamic p63 BSs were driving expression to different tissues, we assigned them to their putatively regulated genes using GREAT[35] and performed an analysis of enrichment of WT expression patterns. Interestingly, we found that genes associated to early p63 BSs were mainly expressed in the neural plate, while genes associated to late p63 BSs were preferentially expressed in the epidermis (Fig. 2c). This result suggests that p63 might be regulating gene expression in a tissue- and stage-dependent manner through different BSs.

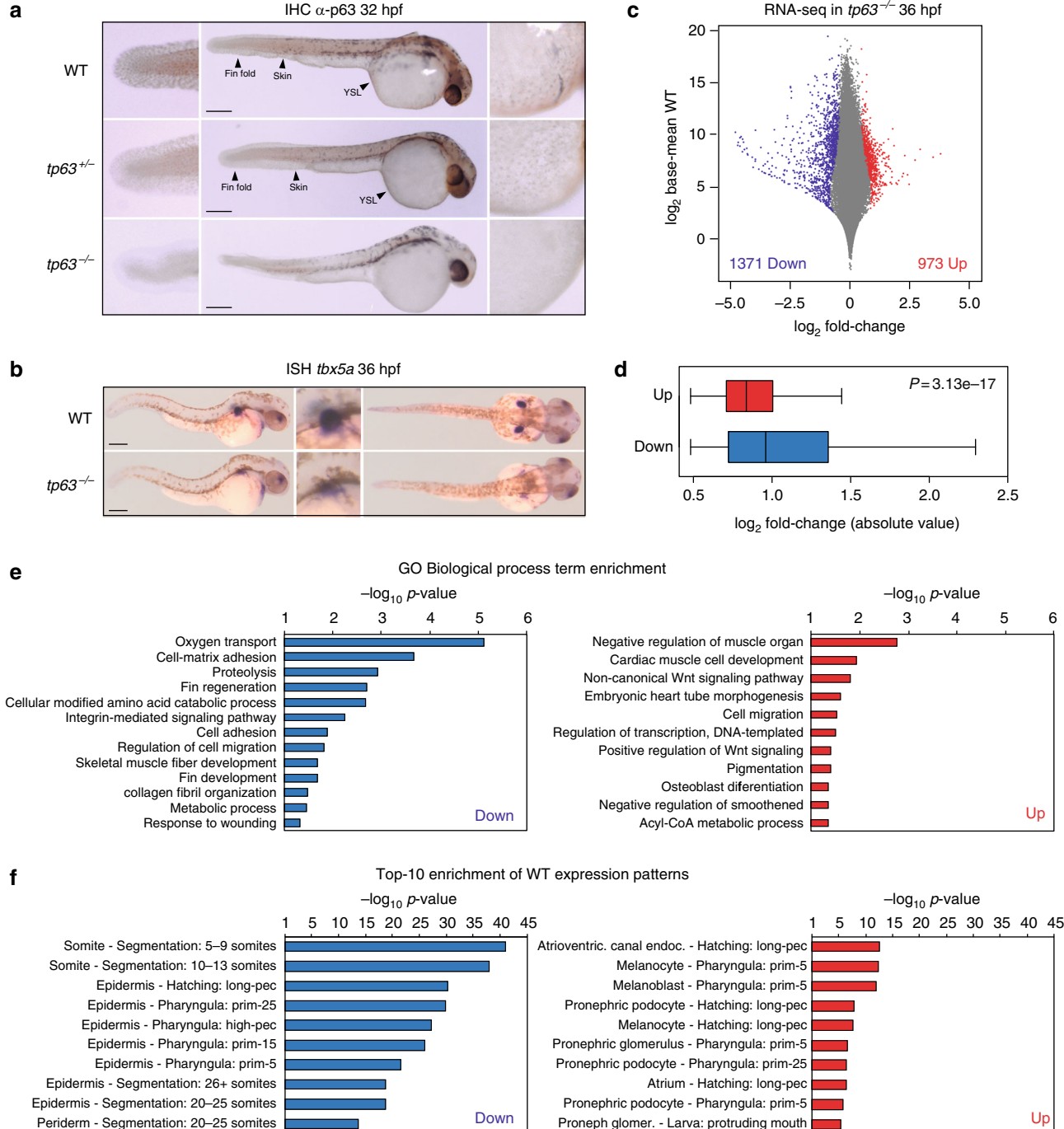

**Fig. 1** Zebrafish *tp63⁻/⁻* mutant embryos show de-regulated epidermal program. **a** Whole-mount embryo immunostaining of p63 in wild-type (WT), heterozygous *tp63⁺/⁻* and homozygous *tp63⁻/⁻* embryos at 32 h post fertilization (hpf) showing the absence of p63 expression in *tp63⁻/⁻* mutants. Left, zoomed lateral view of the tail; middle, lateral view of whole-embryo; right, zoomed lateral view of the yolk. YSL, yolk syncytial layer. **b** Whole-mount in situ hybridization of the *tbx5a* gene in WT and *tp63⁻/⁻* embryos at 36 hpf showing expression in eye, heart and pectoral fin bud, the latter one being reduced in *tp63⁻/⁻* mutant accordingly with the absence of this anatomical structure. Left, lateral view of whole embryo; middle, zoomed lateral view of the pectoral fin bud; right, dorsal view of whole embryo. For **a** and **b**, anterior is to the right. For **a** and **b**, scale bars represent 250 μm. **c** Differential analysis of gene expression between WT and *tp63⁻/⁻* at 36 hpf from RNA-seq ($n = 3$ biological replicates per condition). The $\log_2$ base mean WT transcript expression levels versus the $\log_2$ fold-change of expression are plotted. Transcripts showing a statistically significant differential expression ($p < 0.05$) are highlighted in red (upregulated) or blue (downregulated). **d** Box plot representing the absolute $\log_2$ fold-change of the up- ($n = 973$) and down-regulated ($n = 1,371$) transcripts from (**c**). Center line, median; box limits, upper and lower quartiles; whiskers, 1.5× interquartile range. $p$ value according to the Wilcoxon's rank sum test is shown. Source data are provided as a Source Data file. **e** Gene Ontology (GO) Biological Process term enrichment of the genes corresponding to the up- and down-regulated transcripts from (**c**). **f** Top-10 enrichment of WT expression patterns of the genes corresponding to the up- and down-regulated transcripts from (**c**). For **e** and **f**, the -$\log_{10}$ of $p$-value for each term is shown

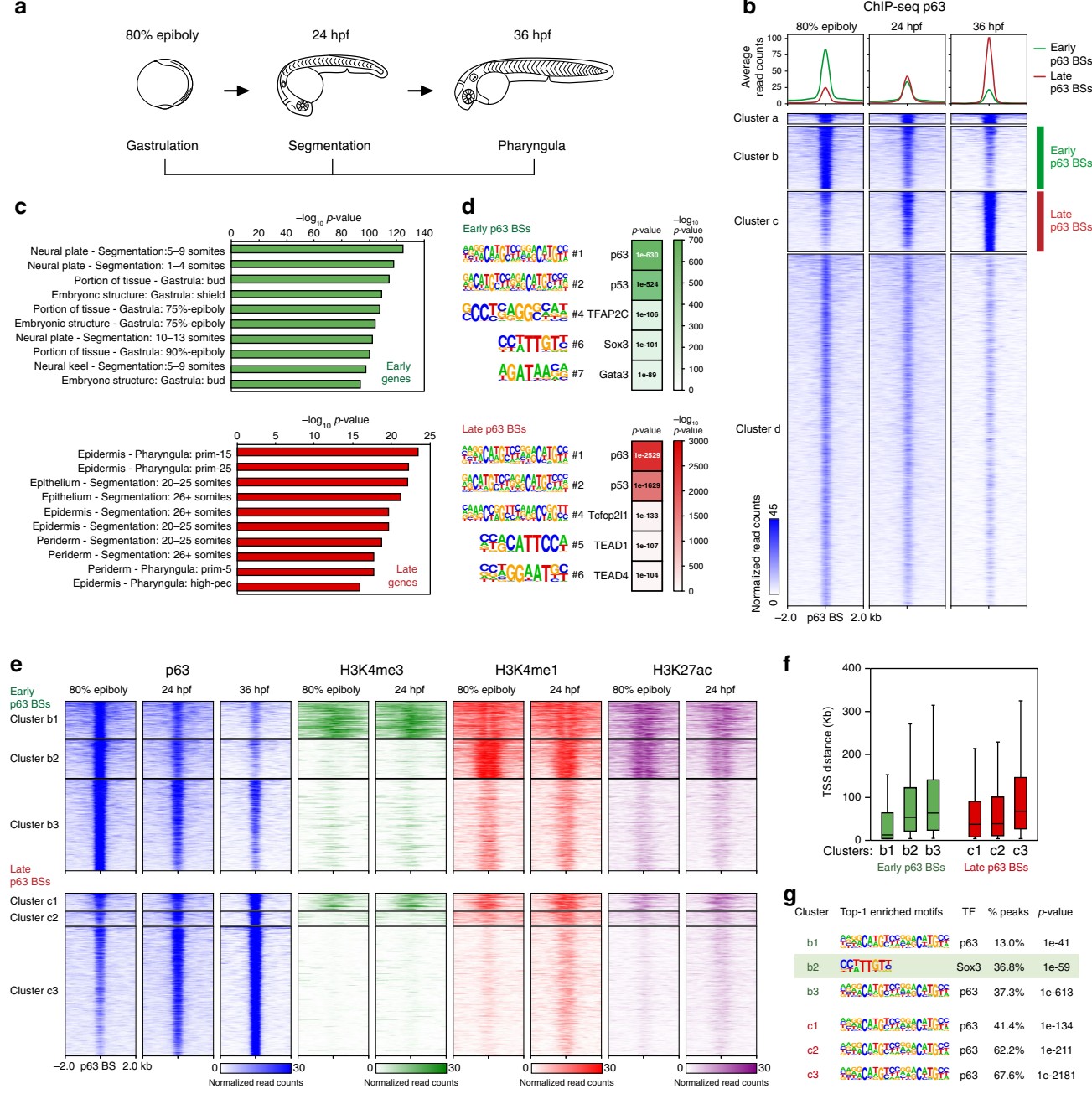

**Fig. 2** Dynamics of p63 binding during development in zebrafish embryos. **a** Picture illustrating zebrafish developmental stages in which most experiments were performed. **b** Heatmaps of the 30,597 p63 binding sites (BSs) obtained from ChIPmentation in 80% of epiboly, 24 and 36 hpf stages (n = 2 biological replicates per stage). Peaks were clustered using k-means clustering, obtaining four groups: cluster a (n = 656), cluster b (n = 3,956), cluster c (n = 3,801) and cluster d (n = 22,184). Average profiles of cluster b, which was used as the group of early p63 BSs, and cluster c, as late p63 BSs, are shown on top. **c** Top-10 enrichment of WT expression patterns of the genes corresponding to early (top) and late (bottom) p63 BSs. The -log₁₀ of p-value for each term is shown. **d** Motif enrichment analysis of the early (top) and late (bottom) p63 BSs. Five representative motifs of the top-10 were chosen. Motif logos are represented with their position in the top-10, the transcription factor (TF) name and the enrichment p-value in a color scale. **e** Heatmaps of the early (top) and late (bottom) p63 BSs clustered according to chromatin marks H3K4me3, H3K4me1 and H3K27ac in 80% of epiboly and 24 hpf. Peaks were clustered using k-means clustering, obtaining three sub-clusters per each group: b1 (n = 875), b2 (n = 912), b3 (n = 2,169), c1 (n = 396), c2 (n = 330) and c3 (n = 3,075). **f** Box plots showing the distance of the peak sub-clusters from (**e**) to the transcription start sites (TSS) of the nearby genes. Center line, median; box limits, upper and lower quartiles; whiskers, 1.5× interquartile range. Source data are provided as a Source Data file. **g** Motif enrichment analyses of the peak sub-clusters from (**e**) representing the most enriched motif in each one, their motif logos, percentage of peaks containing the motif and associated enrichment p-value. See also Supplementary Fig. 1

Next, we performed motif enrichment analyses in clusters b and c. The p63 motif was highly enriched in both clusters, validating the ChIP-seq experiment and peak calling analyses (Fig. 2d, Supplementary Fig. 1), together with other motifs of the

p53 family (i.e., p53, p73). Motifs of already known partners of p63 were found in the early BSs, like GATA family[36] or AP2 family[37] members, but surprisingly also motifs of the Sox family, such as Sox3 and Sox2, which are well known pro-neural TFs. In

the late BSs, we found members of the TEAD family, which have a role in the maintenance of epidermal progenitors[38,39] (Fig. 2d, Supplementary Fig. 1). This temporal segregation of p63 partners suggests a sequential switching of the p63 genetic network during development by regulation of different sets of BSs.

We wondered whether epigenomic features could help to functionally catalog dynamic p63 BSs. Therefore, we clustered early and late BSs according to already available epigenomic marks (H3K4me3, H3K4me1 and H3K27ac) in 80% of epiboly and 24 hpf. This clustering produced different groups of peaks corresponding mainly to promoters, with high signal of all the three epigenomic marks (clusters b1 and c1), and enhancers, enriched in H3K4me1 and H3K27ac signals, but not in H3K4me3 (clusters b2, b3, c2 and c3) (Fig. 2e, f, Supplementary Fig. 1). Motif enrichment analyses in these sub-clusters revealed the p63 motif as the most enriched one in all of them, except for cluster b2, in which the most enriched motif was Sox3 (Fig. 2g, Supplementary Fig. 1). This TF is known to be essential for the establishment of neural fate[40,41], opposite to the p63 function in the developing embryo. Interestingly, cluster b2 represents early enhancers that are already active in dome stage (blastula, 4.3 hpf) (Supplementary Fig. 1) and are also enriched in motifs of other Sox family members and pluripotency factors (Supplementary Fig. 1), suggesting that these enhancers are required for the transition from pluripotency to neural fate. This data, the low proportion of BSs showing the p63 motif in this cluster (11.5%; Supplementary Fig. 1) and the enrichment of neural plate genes associated with early BSs (Fig. 2c), prompted us to explore a possible interaction between p63 and Sox3 that could be important to regulate cell fate during ectoderm specification.

**p63 and Sox3 bind common sites associated to neural genes**. p63 has been shown to co-bind common BSs with Sox2 in cancer cells, where they physically interact[42,43]. Moreover, both Sox2 and Sox3 bind the same DNA sequence in vitro[44] and can substitute for each other during embryonic development[45]. To assess whether p63 and Sox3 share some BSs in zebrafish embryos, we performed ChIPmentation assays for Sox3 in the same developmental stages and compared binding profiles of Sox3 versus p63. Interestingly, a strong overlap was found between both TF BSs, with 18,009 common BSs across the three stages (22.4% of total p63 or Sox3 BSs; Fig. 3a), suggesting that p63 and Sox3 might co-regulate a subset of common BSs. In agreement with the results mentioned above, these overlapping ChIP-seq peaks showed stronger p63 binding at the early stage (Supplementary Fig. 2a) and were specifically associated to genes enriched in neural plate expression (Fig. 3b, Supplementary Fig. 2b), suggesting that both TFs may cooperate in neural plate gene-associated enhancers.

Motif enrichment analysis of these common BSs showed that the most enriched motifs were those of p63 and Sox3, although the percentage of peaks showing the p63 motif was very low (Fig. 3c). This suggests that p63 could be binding these sites either by direct binding to a different motif or by interaction with another factor, as could be the case of the Sox3 motif or Sox3 itself. In order to address this question, we analyzed the distribution of distances between the p63 peaks summits (where the TF is presumably located) to the center of the Sox3 motif at common BSs showing that motif. As a positive control, we compared them with the distances of the p63 summits to the p63 motif in all p63 BSs containing it. Figure 3d shows that p63 summits are indeed centered on the Sox3 motif in these common BSs, suggesting that p63 binds to the Sox3 motif rather than near the Sox3 motif.

To further investigate this possibility, we re-analyzed our ChIPmentation data as standard ATAC-seq data, i.e., mapping

the precise Tn5 cutting sites. This allowed us to analyze the average footprints generated by p63 and Sox3 binding to their motifs in different sets of peaks. Indeed, we found that both p63 and Sox3 binding to their respective motifs resulted in a characteristic footprint different to each other (Fig. 3e). Strikingly, plotting the p63 ChIPmentation signal over the Sox3 motif at common BSs revealed a footprint that was weaker but similar to that of Sox3, while Sox3 binding around the p63 motif did not result in any distinguishable footprint (Fig. 3e). These results reinforce the idea that p63 binds to the Sox3 motif in the BSs shared by both TFs. Although a direct binding of p63 to the Sox3 motif cannot be discarded, the difference with the footprint generated by p63 over its own motif suggests that p63 binding to Sox3 motif could occur through interaction with Sox3, which would act as a docking site for p63 at these sites. This hypothesis is also supported by the fact that p63 and Sox2, which is functionally very similar to Sox3, bind a subset of common BSs and physically interact in cancer cells[42–45].

**p63 and Sox3 are co-expressed at the neural plate border**. The genes encoding p63 and Sox3 are expressed in principle in different domains of the developing embryo. During early embryo development, published single-cell RNA-seq (scRNA-seq) data in zebrafish[46] show that *tp63* is expressed at low levels until 60% of epiboly stage (7 hpf), in which the number of cells showing high *tp63* expression increases and remains higher later on, at least until 30 hpf[12] (Supplementary Fig. 2c). In contrast, *sox3* is highly expressed from very early stages (high stage; 3.3 hpf) and the number of cells showing high expression decreases from 75% of epiboly stage (8 hpf).

The footprint analysis of p63 binding to Sox3 motif suggests that p63 and Sox3 could interact, which would imply that both TFs have to be co-expressed. To check whether there is a population of cells co-expressing them, we first analyzed the scRNA-seq mentioned above[46] to observe the overlaid expression of *tp63* and *sox3* during early development. Analysis of the expression patterns of *sox3* and *tp63* genes showed that, whereas *sox3* expression level was higher in the neuroectoderm and neurogenic placodes branches, *tp63* expression was more restricted to the surface ectoderm (Supplementary Fig. 2d), as expected. Interestingly, when we analyzed specifically the bifurcation between the neural and epidermal branches, we found preferential expression of *sox3* in the neural branch and of *tp63* in the epidermal one, with a population of intermediate cells co-expressing both genes (Fig. 3f).

To confirm this observation at the protein level, we performed double immunofluorescence assays using specific antibodies against p63 and Sox3 in bud stage zebrafish embryos (10 hpf), when ectoderm tissue is being specified. As expected, we observed expression of Sox3 in the neuroectoderm and expression of p63 in the surface ectoderm domains, but also a population of cells in the neural plate border expressing both TFs (Fig. 3g). These results demonstrate that Sox3 and p63 are co-expressed in the neural plate border of zebrafish embryos during early development, where they could be defining the boundary between neuroectoderm and surface ectoderm territories.

**p63 regulates neural expression by limiting Sox3 binding**. In order to assess whether p63 loss affected Sox3 binding, we took advantage of our *tp63*[−/−] mutant and performed ChIPmentation of Sox3 in 36 hpf-embryos, the earliest stage where the mutant phenotype becomes distinguishable (Supplementary Fig. 3a). Differential Sox3 binding analyses of 147,588 Sox3 BSs between WT and mutant embryos revealed 2,976 up-regulated and 941 down-regulated peaks (Fig. 4a, b), suggesting a more prominent role of

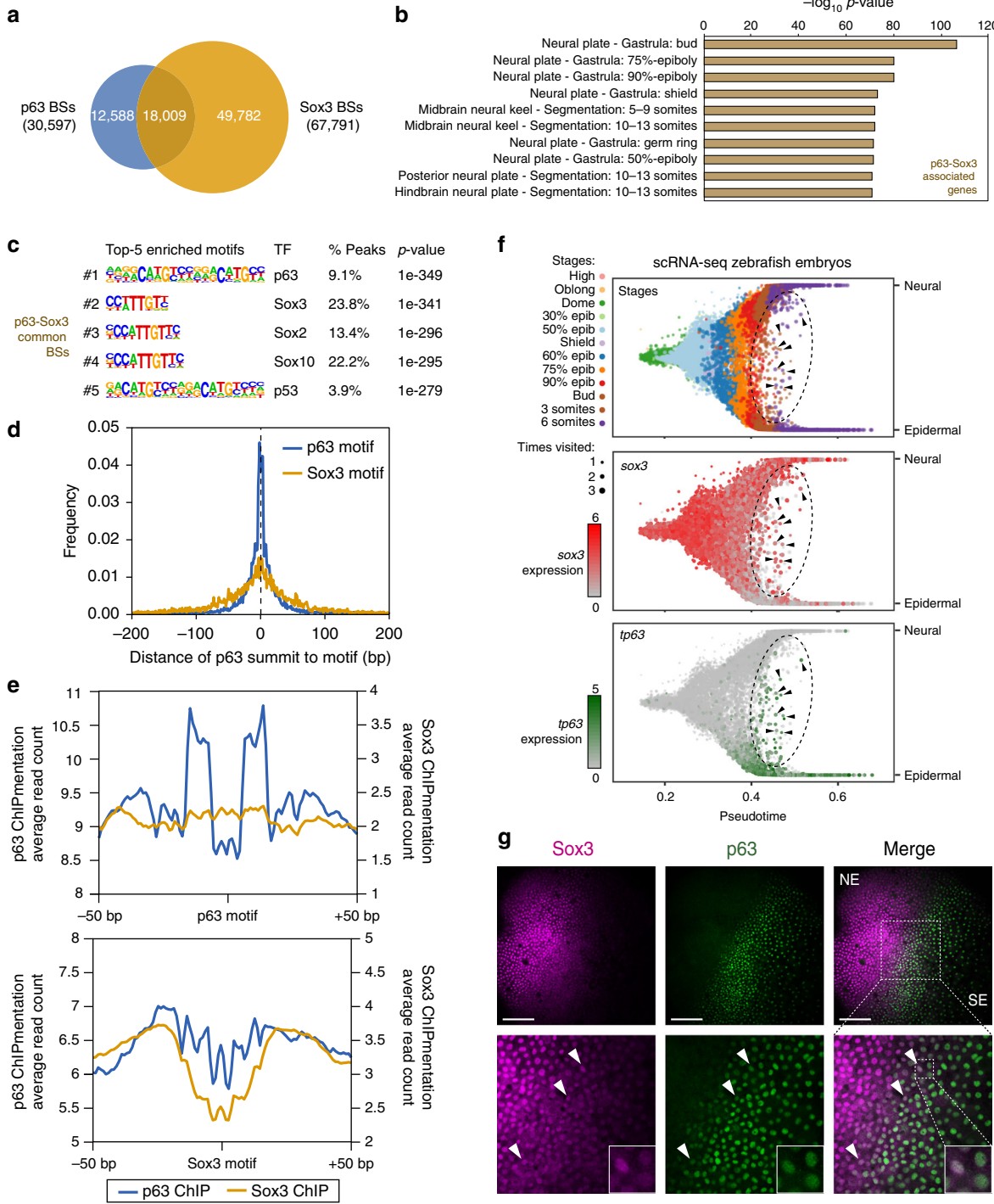

p63 down-regulating Sox3 BSs. 42% of the up-regulated Sox3 BSs in *tp63*$^{-/-}$ mutant overlapped with p63 BSs in WT embryos, but neither of them showed enrichment of the p63 motif (Supplementary Fig. 3b-d). Interestingly, early p63 BSs from our clustering analysis (Fig. 2b) showed enhanced Sox3 signal in the absence of p63, while late p63 BSs did not (Supplementary Fig. 3e), indicating that p63 limits Sox3 binding at early sites enriched in the Sox3 motif. It is worth noting that there is not a detectable change in *sox3* expression in *tp63*$^{-/-}$ mutant embryos (Fig. 4c), suggesting that p63-dependent inhibition of Sox3 binding might occur by impeding directly or indirectly that binding.

To check whether up-regulation of Sox3 BSs in the absence of p63 leads to changes in gene expression, we analyzed the subsets

of genes de-regulated in *tp63*$^{-/-}$ mutant (Fig. 1c) that were also associated to up-, down-regulated or not-changed Sox3 BSs. Strikingly, we found that the expression levels of genes associated to up-regulated Sox3 BSs were also up-regulated in average in *tp63*$^{-/-}$ mutants, while the expression of genes associated to down-regulated or not-changed BSs showed no significant changes compared to all genes (Fig. 4d, e). This suggests that p63 controls the expression of neural genes by regulating Sox3 binding at the nearby enhancers.

**P63 binds to epidermal enhancers before chromatin opening.**
Taking into account that p63 is considered a lineage-determining

**Fig. 3** p63 and Sox3 binding to chromatin and co-expression at the neural plate border. **a** Venn diagram showing the overlap between p63 and Sox3 peaks from ChIPmentation in 80% of epiboly, 24 hpf and 36 hpf embryos ($n = 2$ biological replicates per stage). **b** Top-10 enrichment of WT expression patterns of the genes associated to the common peaks from (**a**). The -$\log_{10}$ p-value for each term is shown. **c** Motif enrichment analysis of the common peaks from (**a**). Top-5 motifs are represented with their motif logos, TF name, percentage of peaks containing the motif and enrichment *p*-value. **d** Distribution of distances from the p63 ChIP-seq summits to either the p63 or Sox3 motifs in total p63 peaks containing the p63 motif ($n = 7,904$) or in common peaks containing the Sox3 motif ($n = 4,286$), respectively, within a 400-bp window centered in the motifs. **e** Footprint analysis of p63 and Sox3 binding to their motifs. Average read counts of Tn5 cutting sites for p63 or Sox3 ChIPmentation signal are plotted centered in either the p63 or Sox3 motifs within a 100-bp window. For p63 signal over p63 motif, total p63 BSs containing the p63 motif ($n = 7,904$) were used. For Sox3 signal over Sox3 motif, total Sox3 BSs containing the Sox3 motif ($n = 20,817$) were used. For p63 signal over Sox3 motif and viceversa, common BSs containing the Sox3 ($n = 4,286$) or the p63 motif ($n = 1,645$), respectively, were used. **f** Branchpoint plots of the ectoderm development from scRNA-seq data in zebrafish early development[46] showing pseudotime (*x*-axis) and random walk visitation preference from the neural to the epidermal domains (*y*-axis). Top, cells colored by developmental stage. Middle and bottom, gene expression of *sox3* and *tp63* in red and green, respectively. An ellipse marks the intermediate region between both domains. Arrowheads point to individual cells co-expressing both genes in this intermediate region. **g** Double whole-mount embryo immunofluorescence of Sox3 (magenta) and p63 (green) in the neural plate border of bud embryos. Arrowheads point to representative nuclei showing co-expression of both markers. NE neuroectoderm, SE surface ectoderm. Scale bars represent 100 μm. See also Supplementary Fig. 2

TF and that a role in regulating chromatin accessibility in epidermal cells has been previously shown[14,16,18], we wondered whether p63 had the same role in our whole-embryo model. To address this, we compared p63 BSs to ATAC-seq data in 80% of epiboly and 24 hpf zebrafish embryos. We called p63 and ATAC-seq peaks in both stages, obtaining a total of 22,583 peaks, and divided them into groups containing a p63 peak, an ATAC-seq peak or both. Then, we plotted the peaks in an alluvial plot to easily visualize transitions between groups across stages (Fig. 5a), excluding the two small groups 2 and 3. Among the different transitions, we noted that the 1,284 peaks of group 4 were characterized by a very low accessibility at 80% of epiboly that became higher at 24 hpf, combined with high p63 binding in both stages (Fig. 5b, Supplementary Fig. 4a). This suggests that p63 could be acting as a pioneer TF in these regions. Pioneer TFs are able to engage non-accessible sites at condensed chromatin (i.e., closed chromatin) and open them later on in a sequential process, displacing nucleosomes to allow binding of non-pioneer TFs or other proteins[25–28]. Since p63 binds to closed chromatin at 80% of epiboly stage that becomes open at 24 hpf, we will thereafter refer to group 4 as pioneered BSs. Note that group 1 of peaks, representing sites bound by p63 in both stages but without being detected as statistically significant ATAC peaks in neither of them, also shows a weak increase in chromatin opening at 24 hpf (Fig. 5a, b), suggesting that it could represent also sites pioneered by p63 in low abundant cell types. Interestingly, the related TF p53 has been described to have pioneer activity[24], and this is likely to be the case for p63 as well.

Epigenomic marks of chromatin activation were consistent with chromatin opening of pioneered BSs, being more active at 24 hpf (Fig. 5c, Supplementary Fig. 5a–c), and motif enrichment analysis of these BSs showed that the p63 consensus sequence was the most enriched motif in this group (Fig. 5d). Furthermore, the genes associated to pioneered BSs were preferentially expressed in the epidermis or related tissues, as is the case of *lama5* (Fig. 5e, f), suggesting that p63 pioneer function may be specific for epidermal gene-associated enhancers. Note that other groups of peaks from the alluvial analyses did not fit these criteria, except for the group 1 that showed similar features to pioneered BSs (Supplementary Fig. 4b–c and 5a–c). Altogether, our results suggest that p63 may be acting as a pioneer TF over a subset of enhancers associated to epidermal genes.

**Pioneer function of p63 promotes epidermal expression**. A pioneer function of p63 over epidermal gene-associated enhancers would imply that the lack of p63 impairs its opening. Indeed, a dependency on p63 for the opening of epidermal gene-associated regulatory regions was recently seen in keratinocytes and embryonic skin[14,16]. To test this hypothesis in our

whole embryo model, we performed ATAC-seq assays in *tp63*$^{-/-}$ embryos at 8 somites stage, soon after ectoderm-derived tissues are specified, and also at 36 hpf, a stage in which neural and epidermal tissues are already formed (Fig. 6a and Supplementary Fig. 5d, e). Comparison of the average chromatin opening at pioneered BSs revealed a decreased ATAC-seq signal in *tp63*$^{-/-}$ mutant compared with control embryos (Fig. 6a). This difference was higher in 8 somites than in 36 hpf stage (Supplementary Fig. 5e), probably due to the dilution of the cell population in a more complex embryo. Importantly, the reduction of ATAC-seq signal in *tp63*$^{-/-}$ mutant was specific of pioneered BSs, although a decrease was also observed in the group 1 of pioneered-like BSs (Fig. 6a). Therefore, these results indicate that p63 is required for the proper establishment of chromatin accessibility in epidermal gene-associated enhancers.

Then we wondered whether this impairment in chromatin opening had any consequence in the expression of associated genes. We observed particular cases in which pioneered p63 BSs down-regulated in *tp63*$^{-/-}$ mutant appeared near epidermal-expressing genes, such as *col18a1a* (Fig. 6b). To test this globally, we compared the fold-change expression of de-regulated genes in *tp63*$^{-/-}$ mutant associated to pioneered BSs with the rest of de-regulated genes in that mutant. Importantly, pioneered BS-associated genes tended to be down-regulated, following the general trend of p63-dependent genes, but this down-regulation was even higher (Fig. 6c). Furthermore, the proportion of down-regulated genes was significantly higher for pioneered BS-associated genes than for the rest of genes, switching from a 57 to a 68% of de-regulated genes (Fig. 6d). Altogether, our results indicate that p63 plays a role as a pioneer TF specifically over epidermal gene-associated enhancers, promoting the expression of epidermal genes.

According to our data, we propose a model in which p63 plays a dual role in vivo during ectoderm specification, with two distinct functions that are temporally, spatially and mechanistically separated (Fig. 6e). During gastrulation, p63 prevents Sox3 binding to early enhancers in the neural plate border, leading to the repression of the neural transcriptomic program. This interaction may limit the expansion of the neural plate into the epidermal domains and promote epidermal specification. On the other hand, p63 binds to non-accessible chromatin at enhancers associated to epidermal genes, promoting their opening by a pioneer function and activating the epidermal transcriptomic program during epidermis differentiation at segmentation and pharyngula stages. The dual role proposed is consistent with previous reports that show activation of the epidermal lineage genetic program by p63 in different systems through chromatin opening[1,14,16,18], and neural expansion during early development for down-regulation of p63 in different animal

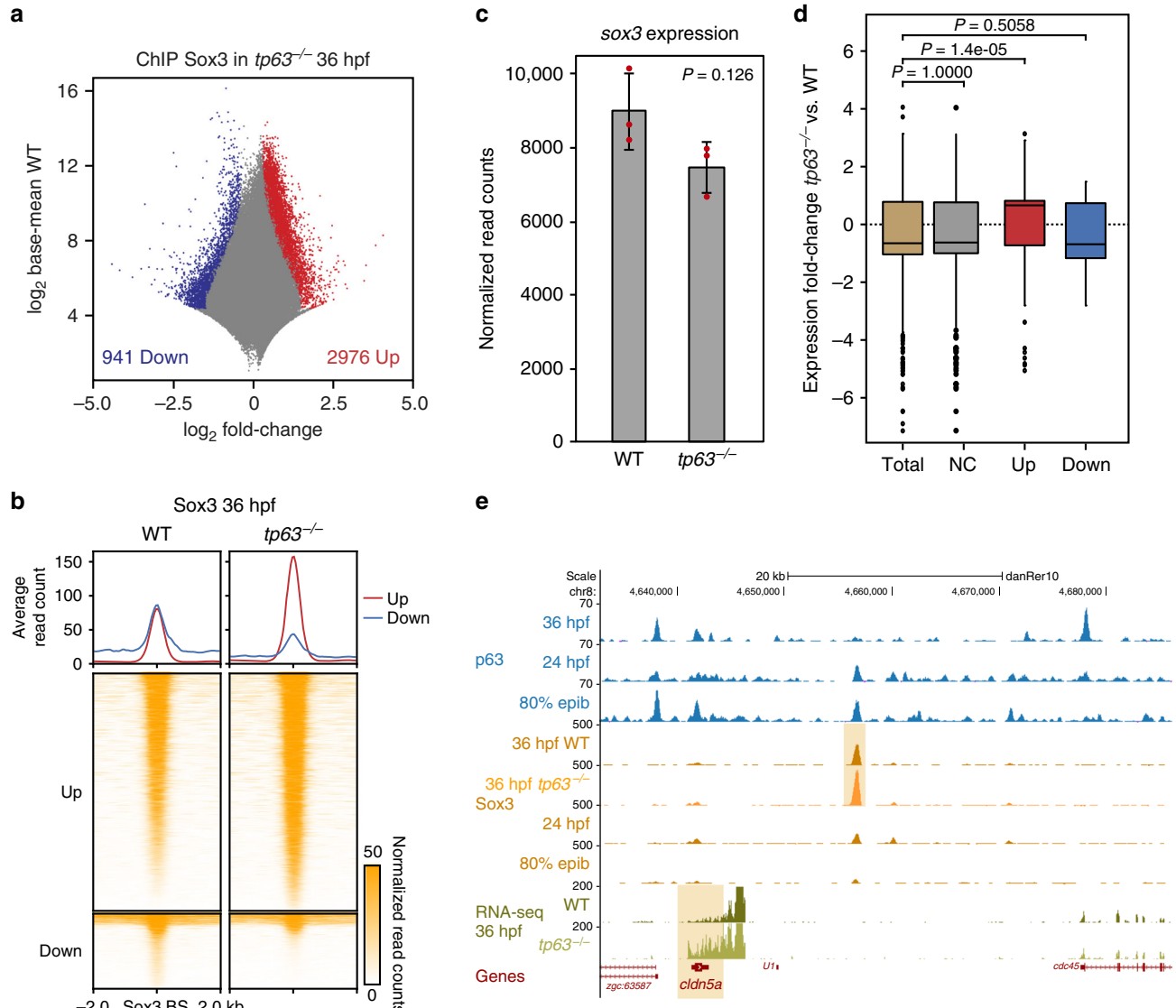

**Fig. 4** p63 regulates Sox3 binding to neural enhancers and the neural program. **a** Differential analysis of Sox3 binding between WT and *tp63*$^{-/-}$ at 36 hpf, calculated in a set of 147,588 Sox3 BSs (FDR < 0.05) from ChIPmentation (*n* = 2 biological replicates per condition). The log$_2$ base mean Sox3 binding levels in the WT versus the log$_2$ fold-change of expression are plotted. Peaks showing a statistically significant differential expression (*p* < 0.01) are highlighted in blue (downregulated) or red (upregulated). **b** Heatmaps and average profiles of the Sox3 BSs at 36 hpf from (**a**) in WT and *tp63*$^{-/-}$ embryos. Peaks were separated according to the differential analysis in up-regulated (Up; n = 2,976) or down-regulated (Down; n = 941) in *tp63*$^{-/-}$ mutant compared to WT. **c** Bar plot with the mean expression of *sox3* gene in WT vs. *tp63*$^{-/-}$ embryos at 36 hpf as measured by RNA-seq (*n* = 3, for each condition). Error bars represent s.d. Individual values are shown as red dots. Non-corrected *p*-value from the differential expression analysis is shown. **d** Box plots showing the expression fold-change in *tp63*$^{-/-}$ embryos of the transcripts associated to Up- (*n* = 222), Down-regulated (*n* = 75) and not changed (NC; *n* = 1,255) Sox3 BSs from (**a**) compared to the total de-regulated transcripts (*n* = 2,344) in that mutant. Center line, median; box limits, upper and lower quartiles; whiskers, 1.5× interquartile range; points, outliers. Statistical significance was assessed using the Wilcoxon's rank sum test. For (**c**) and (**d**), source data are provided as a Source Data file. **e** Genome tracks of p63 ChIP-seq, Sox3 ChIP-seq, and RNA-seq in the indicated developmental stages showing signal intensities in a region of chromosome 8. An up-regulated Sox3 BS in *tp63*$^{-/-}$ mutant and the associated up-regulated gene (*cldn5a*) are highlighted in light orange. The Genes track represents ENSEMBL annotated genes. See also Supplementary Fig. 3

models[11,31]. Further work will be necessary to understand the molecular mechanisms by which p63 inhibits DNA binding of neural TFs, as Sox3, to neural gene-associated enhancers, as well as the specific role of p63 partners, as AP2, GATA or TEAD family members, during epidermal commitment.

## Methods

**Animal experimentation**. Wild-type AB/Tübingen zebrafish strains were maintained and bred under standard conditions. All experiments involving animals conform national and European Community standards for the use of animals in

experimentation and were approved by the Ethical Committees from the University Pablo de Olavide, CSIC and the Andalusian government.

**CRISPR-Cas9 genomic edition**. CRISPR target site to mutate the *tp63* gene was identified using the CRISPRscan online tool[47,48]. A small guide RNA (sgRNA) targeting the exon 3 of the *ΔNp63* variants (exons 5 or 6 of *TAp63* variants) was used with the following target sequence: 5′-GCC CGT ATG ACT GCA CCC TGG GG-3′. The template DNA for sgRNA transcription was generated by PCR using primers sgRNA_tp63_exon3 and sgRNA_universal (Supplementary Table 1) with Phusion DNA polymerase (Thermo Fisher Scientific). sgRNA was in vitro transcribed using the HiScribe T7 Quick High Yield RNA synthesis kit (NEB) using

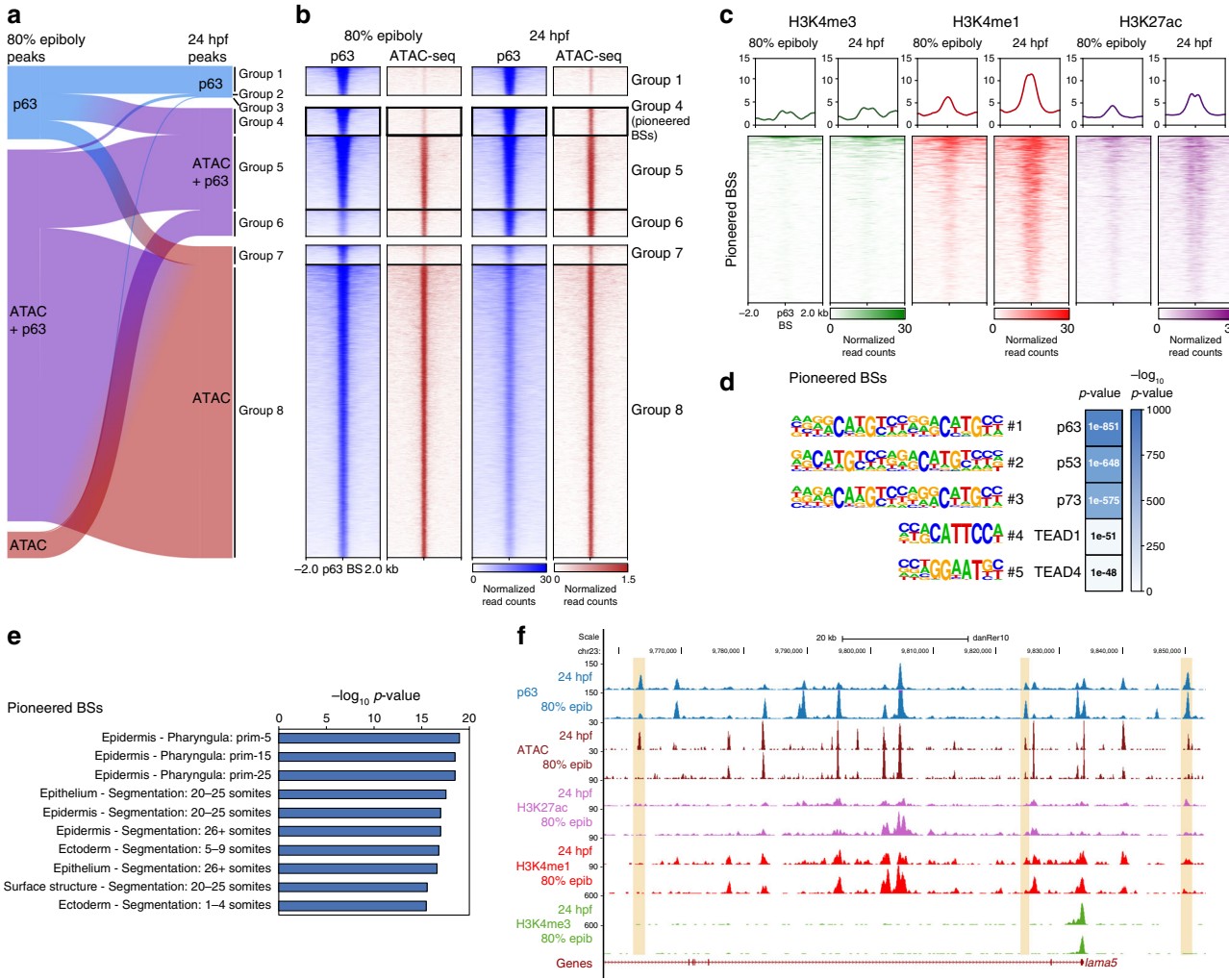

**Fig. 5** p63 binding to epidermal enhancers precedes chromatin opening. **a** Alluvial plot showing the transition between 80% of epiboly and 24 hpf stages of the 22,583 peaks showing p63 binding (blue), chromatin accessibility (ATAC; red) or both (purple). Peaks showing no p63 binding in any of the two analyzed developmental stages were not considered. The transitions lead to eight groups of p63 BSs: group 1 ($n = 332$ peaks), group 2 ($n = 145$ peaks), group 3 ($n = 43$ peaks), group 4 ($n = 1,284$ peaks), group 5 ($n = 3,619$ peaks), group 6 ($n = 1,227$ peaks), group 7 ($n = 896$ peaks) and group 8 ($n = 14,037$). **b** Heatmaps of p63 and ATAC-seq signal in 80% of epiboly and 24 hpf stages for the groups resulting from the alluvial plot transitions in (**a**). Groups 2 and 3 were omitted due to their small number of peaks. **c** Heatmaps and average profiles of H3K4me3, H3K4me1 and H3K27ac in 80% of epiboly and 24 hpf for the group 4 of peaks from (**a**) (pioneered p63 BSs). **d** Motif enrichment analysis of the pioneered p63 BSs. The top-5 motifs are represented with their motif logos, position in the top-5, the TF name and the enrichment p-value in a color scale. **e** Top-10 enrichment of WT expression patterns for the genes corresponding to the pioneered p63 BSs. The -log10 of p-value for each term is shown. **f** Genome tracks of p63 ChIP-seq, ATAC-seq, H3K27ac, H3K4me1, and H3K4me3 in the indicated developmental stages showing signal intensities in a region of chromosome 23. Pioneered p63 BSs associated to the epidermis-expressed gene *lama5* are highlighted in light orange. The Genes track represents ENSEMBL annotated genes. See also Supplementary Fig. 4

75 ng of template, treated with DNase I (NEB) and purified using the RNA Clean and Concentrator kit (Zymo Research).

One-cell stage zebrafish embryos were co-injected with 2–3 nL of a solution containing 300 ng/μL Cas9 protein and 20 ng/μL sgRNA, pre-incubated during 10 min on ice. The CRISPR-Cas9 genomic edition generated a deletion of 4 bp (5′-GTGC-3′; mutation *tp63Δ4*) resulting in a premature STOP codon in the same exon. For screening of the edited genome, genomic DNA was obtained by incubating the samples (whole embryos or adult caudal fin fragments) in TE buffer supplemented with 5% Chelex-100 (BioRad) and 10 μg/mL Proteinase K (Roche) for 1 h (embryos) or 4 h (fins) at 55 °C and 10 min at 95 °C, and then stored at 4 °C. One microliter of the supernatant was used as a template for standard 25-μL PCR reactions using primers tp63-ex3-Fw and tp63-ex3-Rv (Supplementary Table 1). The resulting 338-bp (334-bp for *tp63Δ4* alleles) amplicons were digested with *Pas*I restriction enzyme, since the 4-bp deletion resulted in loss of the *Pas*I site and consequently the enzyme digestion generates 260 and 78-bp bands in WT alleles and a single undigested 334-bp band in *tp63Δ4* alleles. The *tp63Δ4* mutation is

stably maintained in heterozygosis with no apparent phenotypes, as homozygous mutants are embryonic lethal (<3 days).

**Whole-mount embryo immunostaining.** For immunostaining, embryos were fixed overnight at 4 °C with 4% paraformaldehyde, washed in PBT (PBS supplemented with 0.2% Triton-X100) and blocked in this solution with 2% goat serum and 2 mg/mL BSA for 1 h at RT. Then, they were incubated overnight at 4 °C with primary antibody. Primary polyclonal antibodies were anti-Tp63 (used 1:100 dilution, GTX124660 GeneTex) for p63 immunohistochemistry, or anti-p63 (used 1:50 diltuion, 4A4 Abcam) and anti-Sox3 (used 1:100 dilution, GTX132494 GeneTex) for p63 and Sox3 immunofluorescence. After extensive washings with PBT, embryos were incubated overnight at 4 °C with anti-rabbit secondary antibodies conjugated with Hrp (used 1:500 dilution, NA934 Sigma Aldrich) for p63 immunohistochemistry, or goat anti-mouse conjugated with Alexa Fluor 488 (used 1:800 dilution, A28175 Invitrogen) or donkey anti-rabbit conjugated with Alexa Fluor 555 (used 1:500 dilution, A31572 Invitrogen) for p63 and Sox3

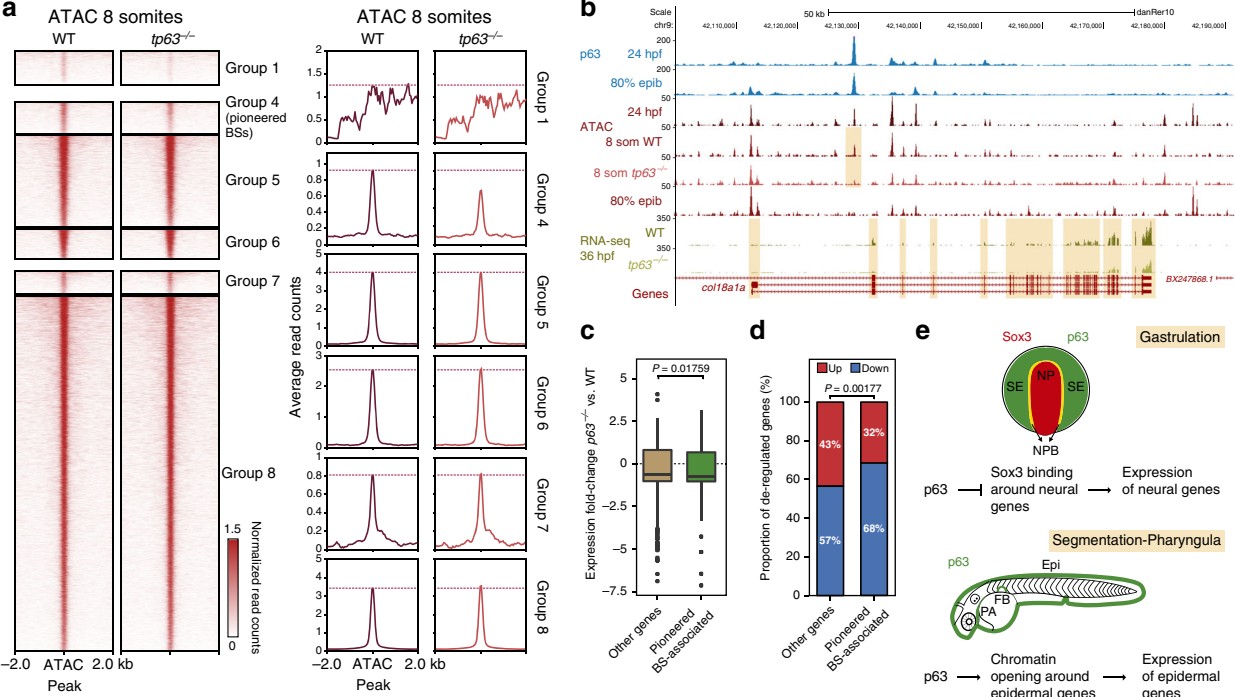

**Fig. 6** p63 acts as a pioneer TF to promote epidermal gene expression. **a** Heatmaps (left) and average profiles (right) of ATAC-seq signal from WT and $tp63^{-/-}$ embryos in 8 somites stage ($n = 2$ biological replicates per condition) for the groups resulting from the alluvial plot transitions from Fig. 5a. A dotted red line represents the WT profile peak in each stage. Groups 2 and 3 were omitted from the analyses due to their small number of peaks. **b** Genome tracks of p63 ChIP-seq, ATAC-seq and RNA-seq in the indicated developmental stages showing signal intensities in a region of chromosome 9. A down-regulated ATAC-seq peak in $tp63^{-/-}$ mutant and the associated down-regulated gene ($col18a1a$) are highlighted in light orange. The Genes track represents ENSEMBL annotated genes. **c** Box plots showing the expression fold-change in $tp63^{-/-}$ embryos of the transcripts associated to group 4 ATAC-seq peaks ($n = 227$) compared to the remaining de-regulated transcripts in that mutant ($n = 2,117$). Center line, median; box limits, upper and lower quartiles; whiskers, 1.5× interquartile range; points, outliers. Statistical significance was assessed using the Wilcoxon's rank sum test. Source data are provided as a Source Data file. **d** Proportion of up- and down-regulated transcripts in $tp63^{-/-}$ embryos associated to ATAC-seq peaks from group 4 compared to the remaining de-regulated transcripts in that mutant. Statistical significance was assessed using the Fisher's exact test. **e** Model proposed to illustrate the dual role of p63 during ectoderm specification. During gastrulation (top), p63 prevents Sox3 binding to enhancers associated to neural genes probably in the neural plate border, where they are co-expressed, resulting in the inhibition of neural expression. During segmentation and pharyngula (bottom), non-accessible chromatin at enhancers associated to epidermal genes and bound by p63 during gastrulation becomes open, triggering an epidermal transcriptomic program that results in the differentiation of epidermis and related structures. NP neural plate, SE surface ectoderm, NPB neural plate border, Epi epidermis, FB fin bud, PA pharyngeal arches. See also Supplementary Fig. 5

immunofluorescence, respectively. For immunohistochemistry, embryos were whole-mounted in agarose and imaged under a MZ-12 dissecting scope (Leica). For immunofluorescence, embryos were flat-mounted and imaged under an SP confocal microscope (Leica).

**Whole-mount embryo in situ hybridization.** Antisense RNA probe against $tbx5a$ gene was prepared from cDNA using digoxigenin (Boehringer Mannheim) as label. Zebrafish embryos were prepared, hybridized and stained using standard protocols[49]. Embryos at 36 hpf stage were fixed in 4% paraformaldehyde overnight, dehydrated in methanol and stored at −20 ℃. All solutions and reagents used were RNAse-free. The embryos were hydrated using decreasing amounts of methanol and finally in PBS-0.1% Tween. Then, they were treated with 10 μg/mL proteinase K for 10' at room temperature and gently washed with PBS-0.1% Tween. In the pre-hybridization step, embryos were kept at 70℃ in the hybridization buffer for at least 1 h. Then, the probe was diluted to 2 ng/μL in hybridization buffer and incubated overnight at 70 ℃ with movement. Pre-heated buffers with decreasing amounts of hybridization buffer (75, 50, 25, and 0%) in 2× SSC solution were used to wash embryos for 10', plus a 30' wash at 70℃ with 0.05× SSC. Then, they were incubated with Blocking Buffer (PBS-0.1% Tween, 2% normal goat serum, 2 mg/mL bovine serum albumin [BSA]) for 1 h, and with antibody anti-digoxigenin (1:5000 in Blocking Buffer) for at least 2 h at room temperature. After this, embryos were washed six times with PBS-0.1% Tween at room temperature and then overnight at 4 ℃. Next day, embryos were washed once more with PBS-0.1% Tween and three times with fresh AP buffer (100 mM Tris-HCl pH 9.5, 50 mM MgCl₂, 100 mM NaCl, 0.1% Tween), followed by signal revelation with NBT/BCIP solution (225 μg/mL NBT, 175 μg/mL BCIP) in multi-well plates in the dark. After tracking the developing signal, the revelation was stopped by washing with PBS-0.1% Tween and fixing with 4% paraformaldehyde. Imaging of the in situ hybridization signal was performed in MZ-12 dissecting scope (Leica).

**RNA-seq.** For total RNA extraction, fifteen WT and fifteen $tp63^{-/-}$ embryos in 36 hpf stage per experiment were collected, manually de-chorionated and suspended in TRIsure (Bioline) with chloroform. Precipitated RNA was then treated with TURBO DNA free kit (Invitrogen). In parallel, DNA was extracted to genotype pooled embryos. Three biological replicates were used for each analyzed line. Illumina libraries were constructed and sequenced in a BGISEQ-500 single-end lane producing around 50 M of 50-bp reads. Reads were aligned to the GRCz10 (danRer10) zebrafish genome assembly using STAR 2.5.3a[50] and counted using the htseq-count tool from the HTSeq 0.8.0 toolkit[51]. Differential gene expression analysis was performed using the DESeq2 1.18.1 package in R 3.4.3[52], setting a corrected $p$ value < 0.05 as the cutoff for statistical significance of the differential expression. Enrichment of GO Biological Process term was calculated using David[53], with a Bonferroni-corrected $p$ value < 0.05 as statistical cutoff. Enrichment of wild-type zebrafish expression patterns was calculated by a custom script (Supplementary Software 1), using a two-tailed hypergeometric test with a Bonferroni-corrected $p$ value < 0.05 as statistical cutoff.

**Single-cell RNA-seq data analyses.** Single-cell RNA-seq data analyses from zebrafish embryos were performed over published data using the URD package[46] in R 3.4.3. Developmental trajectories were calculated between the epidermal and neural branches of the ectoderm specification (69 and 70 nodes, respectively), considering as epidermal the tips 1 and 10, and as neural the tips 2, 8, 16, 19, 50, 56, and 59 (see ref. [46]).

**ATAC-seq**. ATAC-seq assays were performed using standard protocols[54,55], with minor modifications. Briefly, pools of 5 WT or $tp63^{-/-}$ mutant embryos at 36 hpf, or single 8 somites-stage zebrafish embryos coming from $tp63^{+/-}$ parents, were manually de-chorionated. Yolk was dissolved with Ginzburg Ring Finger (55 mM NaCl, 1.8 mM KCl, 1.15 mM NaHCO₃) by pippeting and shaking 5' at 1,100 rpm. Deyolked embryos were collected by centrifugation for 5' at $500 \times g$ 4 °C. Supernatant was removed and embryos washed with PBS. Then, embryos were lysed in 50 μL of Lysis Buffer (10 mM Tris-HCl pH 7.4, 10 mM NaCl, 3 mM MgCl₂, 0.1% NP-40, 1× Roche Complete protease inhibitors cocktail) by pippeting up and down. For 8-somites embryos, the whole cell lysate was used for the next step; for 36-hpf embryos, half of the sample was used to count the cell number in a Neubauer chamber with Hoechst staining, and about 75,000 cells were used. For TAGmentation, cells were centrifuged for 10' at 500 g 4 °C and resuspended in 50 μL of the Transposition Reaction, containing 1.25 μL of Tn5 enzyme and TAGmentation Buffer (10 mM Tris-HCl pH 8.0, 5 mM MgCl2, 10 % w/v dimethylformamide), and incubated for 30' at 37 °C. Immediately after TAGmentation, DNA was purified using the Minelute PCR Purification Kit (Qiagen), and eluted in 20 μL. Before library amplification, purified DNA was used to genotype 8-somites embryos (see above) and two WT and two $tp63^{-/-}$ mutants were selected for deep sequencing. Libraries were generated by PCR amplification using NEBNext High-Fidelity 2X PCR Master Mix (NEB). The resulting library was multiplexed and sequenced in a HiSeq 4000 pair-end lane producing 100 M of 49-bp pair end reads, or in a NextSeq 500 pair-end lane producing 100 M of 51-bp pair end reads. ATAC-seq assays of 80% epiboly and 24 hpf embryos were previously reported[56–58]. Reads were aligned using the GRCz10 (danRer10) zebrafish genome assembly and those pairs separated by more than 2 Kb were removed. The Tn5 cutting site was determined as the position −4 (minus strand) or +5 (plus strand) from each read start, and this position was extended 5 bp in both directions. The IDR framework (idr 0.1 version) was used to obtain high confidence peaks based on replicate information, as previously described[58]. Reads were extended 100 bp only for visualization in the UCSC Genome Browser[59].

For data comparison, all ATAC-seq experiments used (two replicates for each stage, including 8 somites and 36 hpf experiments, as well as previously published 80% epiboly and 24 hpf experiments) were normalized using reads falling into peaks to counteract differences in background levels between experiments and replicates. For this, peaks were first called for each experiment using the IDR framework (see above) and all peaks merged in a single file. Then, the number of reads falling into those peaks was determined for each experiment and the dataset with less reads into peaks was taken as a reference. Finally, the number of total random reads to be taken from each experiment was estimated considering the minimal number of reads into peaks to be taken and the percentage of reads into peaks of each one.

**ChIPmentation**. ChIP-seq of p63 and Sox3 were performed by ChIPmentation, which incorporates Tn5-mediated TAGmentation of immunoprecipitated DNA[34]. Briefly, 3,000 zebrafish embryos in 80% epiboly stage or 1,000 in 24 or 36 hpf stages (120 embryos for Sox3 ChIP in $tp63^{-/-}$ mutants) were dechorionated with 300 μg/mL pronase, fixed for 10 min in 1% paraformaldehyde (in 200 mM phosphate buffer) at RT, quenched for 5 min with 0.125 M glycine, washed in PBS and frozen at −80 °C. Fixed embryos were homogenized in 5 mL cell lysis buffer (10 mM Tris-HCl pH 7.5, 10 mM NaCl, 0.3% NP-40, 1× Roche Complete protease inhibitors cocktail) (2 mL for ChIP in $tp63^{-/-}$ mutants) with a Dounce Homogenizer on ice and centrifuged 5 min 2,300 × g at 4 °C. Pelleted nuclei were resuspended in 1,320 μL of nuclear lysis buffer (50 mM Tris-HCl pH 7.5, 10 mM EDTA, 1% SDS, 1× Roche Complete protease inhibitors cocktail) (333 μL for ChIP in $tp63^{-/-}$ mutants), kept 5 min on ice and diluted with 2,680 μL of ChIP dilution buffer (16.7 mM Tris-HCl pH 7.5, 1.2 mM EDTA, 167 mM NaCl, 0.01% SDS, 1.1% Triton-X100) (667 μL for ChIP in $tp63^{-/-}$ mutants). Then, chromatin was sonicated in a Covaris M220 sonicator (duty cycle 10%, PIP 75 W, 100 cycles/burst, 10 min) or in a Bioruptor 200 sonicator (High intensity, 30-s ON-30-s OFF, 15 min) and centrifuged 5 min 18,000 × g at 4 °C. The recovered supernatant, which contained soluble chromatin, was used for ChIP or frozen at −80 °C after checking the size of the sonicated chromatin. Six 200-μL aliquots of sonicated chromatin (four 250-μL aliquots for ChIP in $tp63^{-/-}$ mutants) were used for each independent ChIP experiment, and each aliquot incubated with 2 μg of anti-p63 (GTX124660 GeneTex) or anti-Sox3 (GTX132494 GeneTex) antibodies and rotated overnight at 4 °C.

Next day, 20 μL of protein G Dynabeads (Invitrogen) per aliquot were washed twice with ChIP dilution buffer and resuspended in 50 μL/aliquot of the same solution. Immunoprecipitated chromatin was then incubated with washed beads for 1 h rotating at 4 °C and washed twice sequentially with wash buffer 1 (20 mM Tris-HCl pH 7.5, 2 mM EDTA, 150 mM NaCl, 1% SDS, 1% Triton-X100), wash buffer 2 (20 mM Tris-HCl pH 7.5, 2 mM EDTA, 500 mM NaCl, 0.1% SDS, 1% Triton-X100), wash buffer 3 (10 mM Tris-HCl pH 7.5, 1 mM EDTA, 250 mM LiCl, 1% NP-40, 1% Na-deoxycholate) and 10 mM Tris-HCl pH 8.0, using a cold magnet (Invitrogen). Then, beads were resuspended in 25 μL of TAGmentation reaction mix (10 mM Tris-HCl pH 8.0, 5 mM MgCl2, 10% w/v dimethylformamide), added 1 μL of Tn5 enzyme and incubated 1 min at 37 °C. TAGmentation reaction was put in the cold magnet and the supernatant discarded. Beads were washed twice again with wash buffer 1 and 1× TE, and eluted twice for 15 min in 100 μL of elution buffer (50 mM NaHCO3 pH 8.8, 1% SDS). The 200 μL of eluted chromatin per aliquot were then decrosslinked by adding 10 μL of 4 M NaCl and 1 μL of 10 mg/mL proteinase K and incubating at 65 °C for 6 h. DNA was purified using Minelute PCR Purification Kit (Qiagen), pooling all aliquots in a single column, and eluted in 20 μL.

Library preparation was performed as previously described for ATAC-seq[54,55] (see above). Libraries were multiplexed and sequenced in a HiSeq 4000 pair-end lane producing around 20 M of 49-bp pair end reads. Reads were aligned to the GRCz10 (danRer10) zebrafish genome assembly using Bowtie[60]. Peaks were called using MACS2 algorithm[61] with an FDR < 0.001. ChIP-seq of H3K4me3, H3K4me1 and H3K27ac were previously reported[62]. For data comparison, ChIP-seq experiments using the same antibody, two replicates of each stage, except for Sox3 in 80% epiboly and 24 hpf, were normalized by taking an equal number of random total reads, which were extended 300 bp for data visualization.

**ChIPmentation and ATAC-seq data analyses**. Conversion of SAM alignment files to BAM was performed using Samtools[63]. Conversion of BAM to BED files, and peak analyses, such as overlaps or merges, were carried out using the Bedtools suite[64]. Conversion of BED to BigWig files was performed using the genomecov tool from Bedtools and the wigToBigWig utility from UCSC. Heatmaps and average profiles of ChIP-seq and ATAC-seq data were generated using computeMatrix, plotHeatmap and plotProfile tools from the Deeptools 2.0 toolkit[65]. Correlation analyses were performed using only the reads falling into peaks ($p < 0.001$) and empirically excluding peaks with abnormally high number of reads. $k$-means clustering was performed when required using Deeptools 2.0 or seqMiner[66] (rank clustering method). Differential analysis of Sox3 peaks in WT versus $tp63^{-/-}$ mutant was performed from the complete set of 147,588 peaks from the four experiments called with MACS2 (FDR < 0.05), using DESeq2 1.18.1 package[52] in R 3.4.3. A corrected $p$ value < 0.01 was set as the cutoff for statistical significance of the differential expression analysis. TF motif enrichment was calculated using the script FindMotifsGenome.pl from Homer software[67], with standard parameters.

For the footprinting analysis, peaks containing the p63 or Sox3 motifs were obtained using the annotatePeaks.pl tool from Homer software. The Tn5 cutting sites were determined as the position −4 (minus strand) or +5 (plus strand) from each read start, and this position was extended 5 bp in both directions, as described above for ATAC-seq. Finally, average read counts around p63 or Sox3 motifs in a 100-bp window centered in the motifs were plotted.

For gene assignment to ChIP and ATAC peaks, coordinates were converted to Zv9 (danRer7) genome using the Liftover tool of the UCSC Genome Browser[59] and assigned to genes using the GREAT tool[35], with the basal plus extension association rule with standard parameters (5 Kb upstream, 1 Kb downstream, 1 Mb maximum extension). Enrichment of wild-type zebrafish expression patterns was calculated by GREAT with standard parameters (significant by both region-based binomial and gene-based hypergeometric test, FDR < 0.05).

**Statistical analyses**. For comparison between expression fold-changes among multiple gene sets, the Kruskal–Wallis rank sum test was first used to test for differences among groups, followed by pairwise comparisons using a two-tailed Wilcoxon's rank sum test. For comparison between expression levels of two gene sets, a two-tailed Wilcoxon's rank sum test was used. In both cases, box plots represent: center line, median; box limits, upper and lower quartiles; whiskers, 1.5× interquartile range; points, outliers. Statistical significance of contingency tables was assessed using the Fisher's exact test.

**Reporting summary**. Further information on research design is available in the Nature Research Reporting Summary linked to this article.

## Data availability

ChIPmentation, ATAC-seq and RNA-seq data generated in this study are available through the Gene Expression Omnibus under accession code GSE123059. Public datasets used in this study are available from GEO: GSE32483 (ChIP-seq of histone modifications), GSE61065 (ATAC-seq 24 hpf) and GSE106428 (ATAC-seq 80% of epiboly). The source data underlying Figs. 1d, 2f, 4c, 4d and 6c are provided as a Source Data File.

## Code availability

Custom code used in this study is available as a Supplementary Software 1. All the data analyses were performed using publicly available software.

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

## Acknowledgements

We thank J.L. Gómez-Skarmeta, J.R. Martínez-Morales and I. Maeso for critical reading of the manuscript; the CABD Fish Facility for technical assistance; S. Muñoz-Galván and A. Carnero for providing reagents; and C3UPO for the HPC support. This work was supported by a grant from the Spanish Ministry of Economy and Competitiveness to J.J.T. (BFU2014-58449-JIN) and an ERC-Advanced grant from the European Research Council (ERC) under the European Union's Horizon 2020 research and innovation program (grant agreement No. 740041). J.M.S.-P. was funded by a postdoctoral contract associated to an Excellence Project from Junta de Andalucía (P12-BIO-396) and by a Juan de la Cierva-Incorporación fellow from the Spanish Ministry of Economy and Competitiveness (IJCI-2016-29884).

## Author contributions

J.J.T. and J.M.S.-P. conceived and designed the project; J.M.S.-P., L.G.-F., and A.N. performed the experiments; J.M.S.-P., R.D.A., and J.J.T. analyzed the data; J.J.T. and J.M.S.-P. wrote the manuscript.

## Additional information

**Competing interests:** The authors declare no competing interests.

**Publisher's note**: 

