## [Peer Review File · Nature Communications]

Reviewers' comments:

Reviewer #1 (Remarks to the Author):

Leveraging ATAC-seq and ChIPmentation, Santos-Pereira et al. demonstrated that p63 binding sites are dynamically regulated during early developmental stages between gastrulation and pharyngula, both promoting epidermal fate and suppressing the neural program. Overall this is an interesting study. It provides an important perspective from whole embryos, which nicely complements the current knowledge mostly generated from isolated cells cultured in plastic dishes.

A few suggestions:

- 1) All the genomic datasets should be deposited to GEO, with GSE# indicated in the paper.
- 2) A minor suggestion for Figure 1: please consider using two different colors to label "up" versus "down" in c, d, e and f.
- 3) The interplay between p63 and sox 3 still remains unclear.
 - 3.1 In Figure 2e, cluster b2 indicates that p63 binds to the sox3 motif. Is p63 directly binding to the sox3 motif, or is p63 binding near the sox3 motif? One possible approach is to analyze the average distance between sox3 motif and the summit of p63 binding in this "cluster b2".
 - 3.2 In Figure 3, both up- and down-regulated sox3 binding sites were identified in p63 KO. How about the regions included in the "cluster b2" as mentioned in Figure 2e? Do these regions have altered sox3 binding with p63 loss?
- 4) Figure 4a and Figure S3. Please add 1 additional panel in both figures to label the cells that express both p63 and SOX3. It is difficult for the reader to identify which cells have both p63 and sox3 when they are labelled separately in two panels.
- 5) Figure 5. The interpretation of Group4 as "pioneered BSs" is confusing. This group has p63 binding at both "epiboly" and "24 hpf" stages, although the signals of both p63 binding and ATAC-seq are higher at 24 hpf. How come the "pioneer function" of p63 was not able to efficiently displace nucleosome at the "epiboly" stage? Would it be possible that p63 binding at these sites is stabilized by other chromatin associated regulators?
- 6) Figure 6.
 - 6.1 Please consider moving FigS5f to Figure 6, as FigS5f is highly relevant.
 - 6.2 Typo in figure legend. Page 23, line #10. should be "e, Model proposed..." instead of "d, Model proposed..."

Reviewer #2 (Remarks to the Author):

In this manuscript, Santos-Pereira and colleagues used zebrafish development as an in vivo model to study the role of p63 in gene and chromatin regulation. Although the concept that p63 is important for controlling the chromatin landscape and may function as a pioneer factor for epidermal keratinocytes is not completely new, previous studies are all performed in cellular modes such as keratinocytes. Authors here showed for the first time during in vivo development, p63 can engage unaccessible chromatin that become open at later stages of development, which is one of the characteristics of pioneer factors. In addition, authors also reported unknown p63 function in repressing the neuronal program. All in all, analyses and data are of high quality, and the study is important for the field. However, several important questions should be addressed, either via more robust analyses or clarification in writing.

Major points:

1. Authors emphasize two distinct roles of p63, early neuronal repression, and late promoting the epidermal fate. This is partially based on the clustering analysis of their ChIP-seq data, early and late clusters. It is known that p63 is expressed at a very low level (if there is at all) at the very early pluripotent stage and its expression is induced when the ectodermal stage is established. At least this is the observation in mouse embryos. Presumably at the 80% epiboly stage (gastrulation), p63 expression is low. It is a bit puzzling when p63 ChIP-seq signals for 80% epiboly stage are higher in Figure 2b, as compared to the signals at later stages, when all clusters are taken into account. What kind of normalized signals are we seeing here? Or are they from z-

scores?

2. Related to the above point, what is the p63 expression level during development, at the mRNA level and at the protein level? Authors stated 'complementary expression patterns of these two TFs (refer to SOX3 and TP63)'. However we do not see p63 and SOX3 expression levels in the manuscript. It is important to show this. This will also help to clarify the following point.

3. The interaction of p63 and SOX3 in regulating gene expression is confusing. Authors suggest that p63 inhibits SOX3 DNA binding without p63 binding to DNA directly. However, they also mention that enhanced SOX3 BSs in p63 mutant are early p63 BSs. These two observations do not seem to be consistent. It is difficult to extrapolate the data how p63 inhibits SOX3 binding, by competing binding sites or by enhancing SOX3 expression? Since that p63 represses the neuronal program during early development is one of the major conclusions in the manuscript, it is important to have better indications on the potential mechanisms.

4. P63 functioning as a pioneer factor is an attractive idea. However in the field, the definitions of pioneer factors are quite different. In Figure 5a, only about half of p63 binding sites that are not accessible (not ATAC signals) at epiboly stage can become accessible later. This suggests that p63 binding is not sufficient to open the chromatin. In this case, can p63 still be called as a pioneer factor? A better definition of pioneer factor and a better evaluation of the role of p63 in this respect will help to clarify.

Minor points:

1. Phenotypes of p63 knockout mouse models are well described. They die after birth, most likely due to dehydration. Authors should briefly discuss the comparison between the zebrafish and the mouse models?

2. For Figure 5a, groups are mentioned but not labeled. Even though groups are indicated in Figure 5b and in supplementary data, it is not clear which groups authors refer to in 5a.

3. Some sentences are not so clearly formulated, e.g. 'we compared the transcriptomic changes in tp63^{-/-} mutants of the genes associated with up-, down-regulated or not-changed Sox3 BSs.' (p6), complicated to understand; what is 'WT expression term enrichment analysis'?

Reviewer #3 (Remarks to the Author):

The manuscript "Pioneer and repressive functions of p63 during embryonic ectoderm specification" by J. M. Santos-Pereira et al. describes the p63 functions in the epidermal differentiation program using tp63-knock-out zebrafish mutants. The authors proposed the model in which p63 plays two distinct roles during ectoderm specification: during early ectoderm specification, the p63 represses the neural fate by impeding Sox3 binding to enhancer in the neural plate border; in later development, it promotes the epidermal fate by a pioneer function over epidermal gene-associated enhancers. The overall findings are quite interesting. Although they stated in their manuscript that "little is known about the mechanisms by which p63 regulates ectoderm specification in living developing embryos", most their findings in this manuscript have been already reported by recent studies in mice embryos and in vitro systems (References 9-13 and 35-36). Therefore, I cannot recommend this manuscript for the publication of Nature Communication. Since RNA-seq, single-cell RNA-seq, and ATAC-seq are useful for zebrafish community, I think they should send this manuscript to the molecular biology journal.

Reviewer #1 (Remarks to the Author):

Leveraging ATAC-seq and ChIPmentation, Santos-Pereira et al. demonstrated that p63 binding sites are dynamically regulated during early developmental stages between gastrulation and pharyngula, both promoting epidermal fate and suppressing the neural program. Overall this is an interesting study. It provides an important perspective from whole embryos, which nicely complements the current knowledge mostly generated from isolated cells cultured in plastic dishes.

Thank you for the positive reception of the manuscript.

A few suggestions:

1) All the genomic datasets should be deposited to GEO, with GSE# indicated in the paper.

Performed as requested.

2) A minor suggestion for Figure 1: please consider using two different colors to label “up” versus “down” in c, d, e and f.

We agree with the reviewer and have marked “up-regulated genes” in red and “down-regulated genes” in blue.

Thank you.

3) The interplay between p63 and sox 3 still remains unclear.

3.1 In Figure 2e, cluster b2 indicates that p63 binds to the sox3 motif. Is p63 directly binding to the sox3 motif, or is p63 binding near the sox3 motif? One possible approach is to analyze the average distance between sox3 motif and the summit of p63 binding in this “cluster b2”.

This is a very good point. Considering that the number of peaks in the cluster b2 containing the binding motif of Sox3 is relatively low (333 peaks, 36.8%), and that there could also be peaks in which these two factors interact out of this group, we decided to perform the suggested analysis in the bigger set of peaks shared by p63 and Sox3 (Fig. 3a). Thus, we have plotted the distribution of distances from the p63 summit to the Sox3 motif in regions harboring peaks of both ChIP-seq experiments and presenting the Sox3 motif (4,286 peaks, 23.8%; see Fig. 3c), and added as a control the distance to the p63 motif in all p63 peaks (Fig. 2b). This new analysis is shown in the new Fig. 3d. As can be seen, p63 binding is centered in the Sox3 motif in those peaks, indicating that p63 binds, directly or indirectly, to that motif.

In addition, we have re-analyzed our ChIPmentation data following an ATAC-seq analysis protocol in order to map Tn5 cut sites and look for footprints of TF binding.

We have found that p63 binds to the Sox3 motif leaving a footprint similar to that generated by Sox3 binding. In contrast, p63 binding to its own motif leaves a completely different footprint, whereas Sox3 does not bind it. This data are also in agreement with the view that p63 binds the Sox3 motif probably through Sox3. These data are shown in new Fig. 3e and commented in the text on page 7 lines 7-30.

3.2 In Figure 3, both up- and down-regulated sox3 binding sites were identified in p63 KO. How about the regions included in the “cluster b2” as mentioned in Figure 2e? Do these regions have altered sox3 binding with p63 loss?

Yes, they do. The early p63 BSs (cluster b), and also cluster b2 in particular, show an up-regulation of Sox3 binding in the absence of p63. However, Sox3 binding is not affected in late p63 BSs (cluster c). We have included this data in new Supplementary Fig. S3e, as well as in the text on page 9 lines 6-9:

“Interestingly, early p63 BSs from our clustering analysis (Fig. 2b) showed enhanced Sox3 signal in the absence of p63, while late p63 BSs did not (Supplementary Fig. 3e), indicating that p63 limits Sox3 binding at early sites enriched in the Sox3 motif.”

4) Figure 4a and Figure S3. Please add 1 additional panel in both figures to label the cells that express both p63 and SOX3. It is difficult for the reader to identify which cells have both p63 and sox3 when they are labelled separately in two panels.

We agree that this could be very useful. Unfortunately, the referred plots contain many overlaid cell layers and are too complex to be combined. On the other hand, it is not computationally possible to make a single color scale based in two different values (i.e. *tp63* and *sox3* expression). To overcome these limitations, we have pointed cells showing co-expression of *tp63* and *sox3* with black arrowheads in the intermediate region between the neural and epidermal branches, in former Fig. 4a (new Fig. 3c).

For former Fig. S3, which is new Fig. S2d, due to difficulty to identify specific isolated cells we have added labels showing the germ layers, the ectoderm sub-layers and the final tips. In any case, the goal of this figure is to show the global expression patterns of *tp63* and *sox3* in all lineages of zebrafish early development.

To clarify, we have added a better explanation in the text on page 8 lines 13-19:

“Analysis of the expression patterns of *sox3* and *tp63* genes showed that, whereas *sox3* expression level was higher in the neuroectoderm and neurogenic placodes branches, *tp63* expression was more restricted to the surface ectoderm (Supplementary Fig. 2d), as expected. Interestingly, when we analyzed specifically the bifurcation between the neural and epidermal branches, we found preferential expression of *sox3* in the neural branch and of *tp63* in the epidermal one, with a population of intermediate cells co-expressing both genes (Fig. 3f).”

5) Figure 5. The interpretation of Group4 as “pioneered BSs” is confusing. This group has p63 binding at both “epiboly” and “24 hpf” stages, although the signals of both p63 binding and ATAC-seq are higher at 24 hpf. How come the “pioneer function” of p63 was not able to efficiently displace nucleosome at the “epiboly” stage? Would it be possible that p63 binding at these sites is stabilized by other chromatin associated regulators?

Pioneer transcription factors have the property to bind their target sites in condensed chromatin and promote its opening. This is a sequential process in which the pioneer factor first engage closed chromatin and then it is opened by itself or by cooperation with other factors and chromatin remodelers (reviewed in refs. 26-28, among others). For that reason, we believe that what we see at 80% of epiboly stage is binding of p63 to condensed chromatin that becomes open later on (at 24 hpf) in a p63-dependent manner. We have clarified this in the text on page 10 lines 3-7:

“Pioneer TFs are able to engage non-accessible sites at condensed chromatin (i.e. closed chromatin) and open them later on in a sequential process, displacing nucleosomes to allow binding of non-pioneer TFs or other proteins²⁵⁻²⁸. Since p63 binds to closed chromatin at 80% of epiboly stage that becomes open at 24 hpf, we will thereafter refer to group 4 as “pioneered BSs”.”

On the other hand, although p63 has been shown to cooperate with chromatin remodelers and epigenetic regulators (refs. 14,15,19-23), whether p63 binding at these sites is stabilized by other factors is not known.

6) Figure 6.

6.1 Please consider moving FigS5f to Figure 6, as FigS5f is highly relevant.

We agree with the importance of former Fig. S5f and, following the reviewer’s suggestion, have moved the experiments performed in 8 somites stage from Fig. S5e and S5e to Figure 6, resulting in new Fig. 6a, leaving in the supplementary the results corresponding to the 36 hpf stage.

Thank you.

6.2 Typo in figure legend. Page 23, line #10. should be “e, Model proposed...” instead of “d, Model proposed...”.

Corrected. Thank you.

Reviewer #2 (Remarks to the Author):

In this manuscript, Santos-Pereira and colleagues used zebrafish development as an in vivo model to study the role of p63 in gene and chromatin regulation. Although the concept that p63 is important for controlling the chromatin landscape and may function as a pioneer factor for epidermal keratinocytes is not completely new, previous studies are all performed in cellular modes such as keratinocytes. Authors here showed for the first time during in vivo development, p63 can engage inaccessible chromatin that become open at later stages of development, which is one of the characteristics of pioneer factors. In addition, authors also reported unknown p63 function in repressing the neuronal program. All in all, analyses and data are of high quality, and the study is important for the field. However, several important questions should be addressed, either via more robust analyses or clarification in writing.

Thank you for the positive reception of the manuscript.

Major points:

1. Authors emphasize two distinct roles of p63, early neuronal repression, and late promoting the epidermal fate. This is partially based on the clustering analysis of their ChIP-seq data, early and late clusters. It is known that p63 is expressed at a very low level (if there is at all) at the very early pluripotent stage and its expression is induced when the ectodermal stage is established. At least this is the observation in mouse embryos. Presumably at the 80% epiboly stage (gastrulation), p63 expression is low. It is a bit puzzling why p63 ChIP-seq signals for 80% epiboly stage are higher in Figure 2b, as compared to the signals at later stages, when all clusters are taken into account. What kind of normalized signals are we seeing here? Or are they from z-scores?

In zebrafish embryos, *tp63* expression at the mRNA level is detected in single cells as soon as the High stage (3.3 hpf), as shown in new Supplementary Fig. 2c (data from ref. 46). From 60% of epiboly (7 hpf) to 6 somites stages (12 hpf), the number of cells expressing *tp63* is notably increased. Expression then remains high at later stages (23, 25 and 30 hpf) (see ref. 12). Therefore, it is not surprising to find a high p63 signal bound to chromatin at 80% of epiboly stage (8.3 hpf). New expression data is cited in the text on page 8 lines 3-8:

“During early embryo development, published single cell RNA-seq (scRNA-seq) data in zebrafish⁴⁶ show that *tp63* is expressed at low levels until 60% of epiboly stage (7 hpf), in which the number of cells showing high *tp63* expression increases and remains higher later on, at least until 30 hpf¹² (Supplementary Fig. S2c). In contrast, *sox3* is highly expressed from very early stages (high stage; 3.3 hpf) and the number of cells showing high expression decreases from 75% of epiboly stage (8 hpf).”

We normalized ChIP-seq signals from different stages by taking the same number of random reads from each experiment. Thus, only differences in the heterogeneity of embryo cell type composition and experimental variability in background signals may account for differences in the average intensity of the called ChIP-seq peaks. In any case, the set of peaks used for the clustering analysis of Fig. 2b comes from the combination of the called peaks in the 3 represented stages, and thus they are not expected to show similar average intensities in all of them. To improve the clarity of the figure, and provided that the cluster “a” is much more intense than the others, we have plotted only the profiles of clusters “b” and “c” (early and late p63 BSs, respectively) in Fig. 2b. It can be seen that the average intensity of the more relevant clusters for this study are similar along the 3 stages. The complete data from the 4 clusters has been moved to new Supplementary Fig. 1b.

2. Related to the above point, what is the p63 expression level during development, at the mRNA level and at the protein level? Authors stated ‘complementary expression patterns of these two TFs (refer to SOX3 and TP63)’. However we do not see p63 and SOX3 expression levels in the manuscript. It is important to show this. This will also help to clarify the following point.

We have included the expression levels of *tp63* and *sox3* genes during zebrafish early development at the mRNA level in new Supplementary Fig. 2c (see above).

Regarding their “complementary” expression pattern, we referred to the fact that both genes are expressed in general in excluding ectodermal lineages. This can be seen at the mRNA level in new Supplementary Fig. 2d, where *sox3* expression in the ectoderm is restricted to the neuroectoderm and neurogenic placodes lineages, while *tp63* is mostly expressed in the surface ectoderm lineage, and also in new Fig. 3g at the protein level (note that in both cases there is a degree of overlap that accounts for p63-Sox3 interaction).

We have clarified this point in the text on page 8 lines 2-3 and 13-16:

“The genes encoding p63 and Sox3 are expressed in principle in different domains of the developing embryo. [...] Analysis of the expression patterns of *sox3* and *tp63* genes showed that, whereas *sox3* expression level was higher in the neuroectoderm and neurogenic placodes branches, *tp63* expression was more restricted to the surface ectoderm (Supplementary Fig. 2d), as expected.”

3. The interaction of p63 and SOX3 in regulating gene expression is confusing. Authors suggest that p63 inhibits SOX3 DNA binding without p63 binding to DNA directly. However, they also mention that enhanced SOX3 BSs in p63 mutant are early p63 BSs. These two observations do not seem to be consistent. It is difficult to extrapolate the data how p63 inhibits SOX3 binding, by competing binding sites or by enhancing SOX3 expression? Since that p63 represses the neuronal program during early

development is one of the major conclusions in the manuscript, it is important to have better indications on the potential mechanisms.

We agree with the reviewer that the referred sentences are confusing and we have removed them for clarity and tried to deepen in the mechanism of interaction between p63 and Sox3. Although we have provided the first evidence showing this relationship, the molecular mechanism of p63-Sox3 interaction is intriguing. Therefore, we have addressed the possibilities of competing BSs and enhanced *sox3* expression.

First, to check how p63 binds at common p63-Sox3 BSs, which show p63 and Sox3 motif enrichment but with a low number of BSs containing the p63 motif (new Fig. 3c), we have analyzed both the distance of the p63 peak summits to the Sox3 motif and the footprint generated by p63 binding around the Sox3 motif at those common sites. These analyses show in new Fig. 3d-e that p63 binding is centered with respect to the Sox3 motif and it generates a footprint similar to that left by Sox3 binding, while Sox3 does not leave a footprint around the p63 motif. This indicates that p63 is able to bind the Sox3 motif, directly or indirectly, but Sox3 does not bind the p63 motif. Although a putative affinity of p63 binding the Sox3 motif cannot be discarded, it is more likely that p63 binds it through physical interaction with Sox3 or other factors. Indeed, p63 co-bind a subset of sites and physically interacts with the highly related Sox2 TF in squamous cell carcinomas (refs. 42-43).

On the other hand, we have also observed that the expression levels of the *sox3* gene are not changed in the absence of p63 (see new Fig. 4c), indicating that differential Sox3 binding upon p63 loss is not an indirect effect of differential gene expression. This supports our hypothesis that p63 inhibits the binding of Sox3 to chromatin to block the neural program and limit the expansion of the neuroectoderm.

The new data have been explained in the text on page 6 lines 7-30, and on page 9 lines 9-11:

“It is worth noting that there is not a detectable change in *sox3* expression in *tp63*^{-/-} mutant embryos (Fig. 4c), suggesting that p63-dependent inhibition of Sox3 binding might occur by impeding directly or indirectly that binding.”

4. P63 functioning as a pioneer factor is an attractive idea. However in the field, the definitions of pioneer factors are quite different. In Figure 5a, only about half of p63 binding sites that are not accessible (not ATAC signals) at epiboly stage can become accessible later. This suggests that p63 binding is not sufficient to open the chromatin. In this case, can p63 still be called as a pioneer factor? A better definition of pioneer factor and a better evaluation of the role of p63 in this respect will help to clarify.

As the reviewer says, we found that only about half of the p63 BSs at non-accessible chromatin at 80% of epiboly stage (groups 1 and 4) were detected as open chromatin (i.e. ATAC peaks) at 24 hpf stage (group 4). This could be explained by two reasons:

1- Group 1 of peaks could correspond to a small cell population that, although showing chromatin opening at 24 hpf, this is diluted in the whole embryo leading to a low ATAC signal. This would imply considering group 1 also as “pioneered BSs”. Indeed, a slight chromatin opening can be seen by eye for group 1 at 24 hpf (Fig. 5a), as well as a slight dependency on p63 for this opening (new Fig. 6a). In addition, group 1 BSs showed a high enrichment of the p63 motif and association to genes expressed in the epidermis (Supplementary Fig. 4b and c), as well as a slight enhancer activation (H3K4me1 and H3K27ac in Supplementary Fig. 5b), features that are shared with group 4 but with any other group. However, we decided to be conservative and exclude group 1 from the “pioneered” p63 BSs.

2- Chromatin opening by p63 requires additional factors that are present in the context of group 4 BSs but not in group 1. Indeed, p63 has been reported to cooperate with chromatin remodelers such as BAF and Brg1 (refs. 14,19). In addition, other pioneer TFs are assisted by chromatin remodelers in order to fully open chromatin, including Oct4 and Gata3 (King, eLife 2017; Takaku, Genome Biol 2016). Therefore, the requirement of chromatin remodelers to open chromatin would not imply that the TF is not a pioneer.

We have clarified this in the text on page 10 lines 7-10 and page 11 lines 3-5:

“Note that group 1 of peaks, representing sites bound by p63 in both stages but without being detected as statistically significant ATAC peaks in neither of them, also shows a weak increase in chromatin opening at 24 hpf (Fig. 5a-b), suggesting that it could represent also sites pioneered by p63 in low abundant cell types. [...] Importantly, the reduction of ATAC-seq signal in *tp63*^{-/-} mutant was specific of pioneered BSs, although a decrease was also observed in the group 1 of pioneered-like BSs (Fig. 6a).”

Minor points:

1. Phenotypes of p63 knockout mouse models are well described. They die after birth, most likely due to dehydration. Authors should briefly discuss the comparison between the zebrafish and the mouse models?

We have compared the phenotypes of zebrafish and mouse *tp63*^{-/-} models on page 4 lines 16-26:

“Mutant animals died just after hatching, between 40 and 50 hours post-fertilization (hpf), and from 36 hpf they showed defects in ectoderm-derived structures, including skin, pectoral fin buds and the fin fold, as reported previously in humans^{4,6,7}. In *tp63*^{-/-} embryos, *in situ* hybridization of *tbx5a* shows the formation of the mesodermal anlage

that will generate the fin bud, but the apical ectodermal ridge (AER) is not formed and therefore the appendage does not grow (Fig. 1b). These phenotypes are similar to those described for *trp63* knockout mice, which die after birth due to dehydration and show craniofacial abnormalities, limb truncations as a result of failure of the AER to differentiate, and absence of epidermis and related appendages, including hair follicles, teeth and mammary glands^{9,10}. Zebrafish embryos knocked-down for *tp63* also showed similar but milder phenotypes^{11,12}.”

2. For Figure 5a, groups are mentioned but not labeled. Even though groups are indicated in Figure 5b and in supplementary data, it is not clear which groups authors refer to in 5a.

Performed as requested. Thank you.

3. Some sentences are not so clearly formulated, e.g. ‘we compared the transcriptomic changes in *tp63*^{-/-} mutants of the genes associated with up-, down-regulated or not-changed Sox3 BSs.’ (p6), complicated to understand; what is ‘WT expression term enrichment analysis’?

We have clarified the first point on page 9 lines 12-14:

“To check whether up-regulation of Sox3 BSs in the absence of p63 leads to changes in gene expression, we analyzed the subsets of genes de-regulated in *tp63*^{-/-} mutant (Fig. 1c) that were also associated to up-, down-regulated or not-changed Sox3 BSs.”

Regarding “WT expression term enrichment analyses”, we agree that the term is confusing and have changed it by “enrichment of WT expression patterns” throughout the manuscript.

Thank you.

Reviewer #3 (Remarks to the Author):

The manuscript “Pioneer and repressive functions of p63 during embryonic ectoderm specification” by J. M. Santos-Pereira et al. describes the p63 functions in the epidermal differentiation program using tp63-knock-out zebrafish mutants. The authors proposed the model in which p63 plays two distinct roles during ectoderm specification: during early ectoderm specification, the p63 represses the neural fate by impeding Sox3 binding to enhancer in the neural plate border; in later development, it promotes the epidermal fate by a pioneer function over epidermal gene-associated enhancers. The overall findings are quite interesting. Although they stated in their manuscript that “little is known about the mechanisms by which p63 regulates ectoderm specification in living developing embryos”, most their findings in this manuscript have been already reported by recent studies in mice embryos and *in vitro* systems (References 9-13 and 35-36). Therefore, I cannot recommend this manuscript for the publication of Nature Communication. Since RNA-seq, single-cell RNA-seq, and ATAC-seq are useful for zebrafish community, I think they should send this manuscript to the molecular biology journal.

We are sorry that the reviewer does not agree with the novelty of our study. However, we believe that our model is very useful to understand how p63 works in a whole embryo context, in which many cell lineages are being specified and interacting with each other. Previous works on the role of p63 regulating the chromatin landscape have been performed in *in vitro* systems (refs. 13-15,18) or in the dorsal skin of mice embryos (ref. 16). They clearly showed that p63 is required to regulate the accessibility of enhancers associated with epidermal genes, but the mechanism remained unclear.

Our work in whole zebrafish embryos along development not only supports this view, but also provides the first ChIP-seq analyses of p63 in whole embryos, which together with ATAC-seq and RNA-seq has allowed us to see: (1) the first evidence of p63 binding to non-accessible chromatin and promoting its opening and the expression of the associated epidermal genes in a dynamic manner; and (2) an unanticipated role of p63 in regulating the neural fate during early development by its interaction at the chromatin level with the pro-neural TF Sox3.

Therefore, we believe that this is a powerful model able to complement the knowledge based in cell lines, stem cells or dissected embryos in which broader interactions and functions of developmental regulators can be detected beyond these very specific contexts.

REVIEWERS' COMMENTS:

Reviewer #1 (Remarks to the Author):

The depth of the manuscript has been significantly improved. All the my concerns have been addressed.

Reviewer #2 (Remarks to the Author):

Authors have done extensive new analyses and comprehensively addressed my questions. The re-analysis of ChIPmentation data using ATAC-seq method is a clever way to identify associated transcription factors. It would be good if authors elaborate a bit more in Methods on how this analysis was performed. E.g. authors state 'i.e. mapping the precise Tn5 cutting sites from our reads'. But the Tn5 cutting sites are not present in ChIPmentation data. Did authors used a window around the summit of ChIP peaks? In addition, I would suggest that authors check the complete manuscript carefully, and rephrase some sentences to make it more clear, e.g. 'p63-null mice have been shown to up-regulate genes required for mesoderm development' (page 3 line 56).

REVIEWERS' COMMENTS:

Reviewer #1 (Remarks to the Author):

The depth of the manuscript has been significantly improved. All the my concerns have been addressed.

Thank you for the positive reception of our revised manuscript and for revising it. Your comments were very helpful to improve the quality of the paper.

Reviewer #2 (Remarks to the Author):

Authors have done extensive new analyses and comprehensively addressed my questions. The re-analysis of ChIPmentation data using ATAC-seq method is a clever way to identify associated transcription factors. It would be good if authors elaborate a bit more in Methods on how this analysis was performed. E.g. authors state 'i.e. mapping the precise Tn5 cutting sites from our reads'. But the Tn5 cutting sites are not present in ChIPmentation data. Did authors used a window around the summit of ChIP peaks? In addition, I would suggest that authors check the complete manuscript carefully, and rephrase some sentences to make it more clear, e.g. 'p63-null mice have been shown to up-regulate genes required for mesoderm development' (page 3 line 56).

Thank you for the positive reception of our revised manuscript and for revising it. We found your suggestions very constructive and they contributed to improve the paper.

Regarding the footprinting analysis, we have included a detailed explanation of how it was performed in the Methods section:

"For the footprinting analysis, peaks containing the p63 or Sox3 motifs were obtained using the annotatePeaks.pl tool from Homer software. The Tn5 cutting sites were determined as the position -4 (minus strand) or +5 (plus strand) from each read start, and this position was extended 5 bp in both directions, as described above for ATAC-seq. Finally, average read counts around p63 or Sox3 motifs in a 100-bp window centered in the motif were plotted."

On the other hand, we have carefully revised the manuscript and corrected unclear sentences as the one indicated by the referee:

"p63-null mice show an up-regulation of genes required for mesoderm development."

Thank you for bringing our attention to that.